# In-and-Out: Algorithmic Diffusion for Sampling Convex Bodies

**Yunbum Kook**
School of Computer Science
Georgia Institute of Technology
yb.kook@gatech.edu

**Santosh S. Vempala**
School of Computer Science
Georgia Institute of Technology
vempala@gatech.edu

**Matthew S. Zhang**
Department of Computer Science,
University of Toronto,
matthew.zhang@mail.utoronto.ca

## Abstract

We present a new random walk for uniformly sampling high-dimensional convex bodies. It achieves state-of-the-art runtime complexity with stronger guarantees on the output than previously known, namely in Rényi divergence (which implies TV, $\mathcal{W}_2$, KL, $\chi^2$). The proof departs from known approaches for polytime algorithms for the problem — we utilize a stochastic diffusion perspective to show contraction to the target distribution with the rate of convergence determined by functional isoperimetric constants of the stationary density.

## 1 Introduction

Generating random samples from a high-dimensional convex body is a basic algorithmic problem with myriad connections and applications. The core of the celebrated result of [1], giving a randomized polynomial-time algorithm for computing the volume of a convex body, was the first polynomial-time algorithm for uniformly sampling convex bodies. In the decades since, the study of sampling has led to a long series of improvements in its algorithmic complexity [2, 3, 4, 5, 6], often based on uncovering new mathematical/geometric structure, establishing connections to other fields (e.g., functional analysis, matrix concentration) and developing new tools for proving isoperimetric inequalities and analyzing Markov chains. With the proliferation of data and the increasing importance of machine learning, sampling has also become an essential algorithmic tool, with applications needing samplers in very high dimension, e.g., scientific computing [7, 8, 9], systems biology [10, 11], differential privacy [12, 13] and machine learning [14, 15].

Samplers for convex bodies are based on Markov chains (see §5 for a summary). Their analysis is based on bounding the *conductance* of the associated Markov chain, which in turn bounds the mixing rate. Analyzing the conductance requires combining delicate geometric arguments with (Cheeger) isoperimetric inequalities for convex bodies. An archetypal example of the latter is the following: for any measurable partition $S_1, S_2, S_3$ of a convex body $\mathcal{K} \subset \mathbb{R}^d$, we have

$$\text{vol}(S_3) \geq \tfrac{d(S_1, S_2)}{C_\mathcal{K}} \min\{\text{vol}(S_1), \text{vol}(S_2)\},$$

where $d(\cdot, \cdot)$ is the (minimum) Euclidean distance, and $C_\mathcal{K}$ is an isoperimetric constant of the uniform distribution over $\mathcal{K}$. (The KLS conjecture posits that $C_\mathcal{K} = \mathcal{O}(1)$ for any convex body $\mathcal{K}$ in *isotropic position*, i.e., under the normalization that a random point from $\mathcal{K}$ has identity covariance). The coefficient $C_\mathcal{K}^2$ is bounded by the Poincaré constant of the uniform distribution over $\mathcal{K}$ (and they are

in fact asymptotically equal). The classical proof of conductance uses geometric properties of the random walk at hand to reduce the analysis to a suitable isoperimetric inequality (see e.g., [3, 16]). The end result is a guarantee on the number of steps after which the total variation distance (TV distance) between the current distribution and the target is bounded by a desired error parameter. This framework has been widely used and effective in analyzing an array of candidate samplers, e.g., Ball walk [4], Hit-and-Run [17, 5], Riemannian Hamiltonian Monte Carlo [18] etc.

One successful approach, studied intensively over the past decade, is based on *diffusion*. The basic idea is to first analyze a continuous-time diffusion process, typically modeled by a *stochastic differential equation* (SDE), and then show that a suitable time-discretization of the process, sometimes together with a Metropolis filter, converges to the desired distribution efficiently. A major success along this line is the Unadjusted Langevin Algorithm and its variants, studied first in [19]. These algorithms have strong guarantees for sampling "nice" distributions [20, 21, 22, 23], such as ones that are strongly log-concave, or more generally distributions satisfying isoperimetric inequalities, while also obeying some smoothness conditions. The analysis of these algorithms is markedly different from the conductance approach, and typically yields guarantees in stronger metrics such as the KL-divergence.

Our starting point is the following question:

> *Can diffusion-based approaches be used for the problem of sampling convex bodies?*

Despite remarkable progress, thus far, constrained sampling problems have evaded the diffusion approach, except as a high-level analogy (e.g., the Ball walk can be viewed as a discretization of Brownian motion, but this alone does not suggest a route for analysis) or with significantly worse convergence rates (e.g., [24, 25]).

**Contributions.** Our main finding is a simple diffusion-based algorithm that can be mapped to a stochastic process (and, importantly, to a pair of forward and backward processes), such that the rate of convergence is bounded directly by an appropriate functional inequality for the target distribution. As a consequence, for the first time, we obtain clean end-to-end guarantees in the Rényi divergence (which implies guarantees in other well known quantities such as $\mathcal{W}_2, \mathsf{TV}, \mathsf{KL}, \chi^2$ etc.), while giving state-of-the-art runtime complexity for sampling convex bodies (e.g., Ball walk or Speedy walk [3, 4]). Besides being a stronger guarantee on the output, Rényi divergence is of particular interest for differential privacy [13]. Perhaps most interesting is that our proof approach is completely different from prior work on convex body sampling. In summary,

- The guarantees hold for the $q$-Rényi divergences while matching the rates of previous work (prior work only had guarantees in the TV distance).
- The analysis is simple, modular, and easily extendable to several other settings.

**Organization.** In §2, we provide some relevant notions for understanding our results. We then proceed to outline our algorithm in §3. The algorithmic guarantees are provided in §4, in which we also outline our proof and compare it with the analysis of Ball walk, Speedy walk. Lastly, we provide a detailed survey of the relevant literature in §5 before concluding.

## 2 Preliminaries

Unless otherwise specified, we will use $\|\cdot\|$ for the 2-norm on $\mathbb{R}^d$. We write $a = \mathcal{O}(b), a \lesssim b$ to mean that $a \leq cb$ for some universal constant $c > 0$. Similarly, $a \gtrsim b, a = \Omega(b)$ for $a \geq cb$, while $a = \Theta(b)$ means $a \lesssim b, b \lesssim a$ simultaneously. We will also use $a = \widetilde{\mathcal{O}}(b)$ to denote $a = \mathcal{O}(b \operatorname{polylog}(b))$. Lastly, we will use measure and density interchangeably when there is no confusion.

To quantify the convergence rate, we introduce some common divergences between distributions.

**Definition 1** (Distance and divergence). For two measures $\mu, \nu$ on $\mathbb{R}^d$, the *total variation* distance between them is defined by

$$\|\mu - \nu\|_{\mathsf{TV}} := \sup_{B \in \mathcal{F}} |\mu(B) - \nu(B)|,$$

where $\mathcal{F}$ is the collection of all measurable subsets of $\mathbb{R}^d$. The 2-*Wasserstein distance* is given by

$$\mathcal{W}_2^2(\mu, \nu) := \inf_{\gamma \in \Gamma(\mu, \nu)} \mathbb{E}_{(X,Y) \sim \gamma}[\|X - Y\|^2],$$

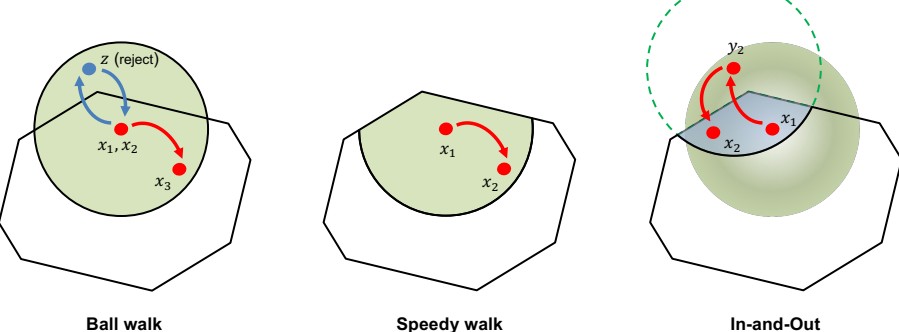

Figure 1: Description of uniform samplers: (i) Ball walk: proposes a uniform random point $z$ from $B_\delta(x_1)$, but $z \notin \mathcal{K}$ so it stays at $x_1 = x_2$. (ii) Speedy walk: moves to $x_2$ drawn uniformly at random from $\mathcal{K} \cap B_\delta(x_1)$. (iii) In-and-Out: first moves to $y_2$ obtained by taking a Gaussian step from $x_1$, and then to $x_2$ obtained by sampling the truncated Gaussian $\mathcal{N}(y_2, h I_d)|_{\mathcal{K}}$.

where $\Gamma$ is the set of all couplings between $\mu, \nu$. Next, we define the *f-divergence* of $\mu$ towards $\nu$ with $\mu \ll \nu$ (i.e., $\mu$ is absolutely continuous with respect to $\nu$) as, for some convex function $f : \mathbb{R}_+ \to \mathbb{R}$ with $f(1) = 0$ and $f'(\infty) = \infty$,

$$D_f(\mu \,\|\, \nu) := \int f\left(\tfrac{\mathrm{d}\mu}{\mathrm{d}\nu}\right) \mathrm{d}\nu \,.$$

The KL-*divergence* arises when taking $f(x) = x \log x$, the $\chi^q$-*divergence* when taking $f(x) = x^q - 1$, and the *q-Rényi divergence* is given by

$$\mathcal{R}_q(\mu \,\|\, \nu) := \tfrac{1}{q-1} \log\left(\chi^q(\mu \,\|\, \nu) + 1\right) \,.$$

**Definition 2.** We say that a probability measure $\nu$ on $\mathbb{R}^d$ satisfies a *Poincaré inequality* (PI) with parameter $C_{\mathsf{PI}}(\nu)$ if for all smooth functions $f : \mathbb{R}^d \to \mathbb{R}$,

$$\mathrm{Var}_\nu f \leq C_{\mathsf{PI}}(\nu) \, \mathbb{E}_\nu[\|\nabla f\|^2] \,, \tag{PI}$$

where $\mathrm{Var}_\nu f := \mathbb{E}_\nu[|f - \mathbb{E}_\nu f|^2]$.

The Poincaré inequality is implied by the log-Sobolev inequality.

**Definition 3.** We say that a probability measure $\nu$ on $\mathbb{R}^d$ satisfies a *log-Sobolev inequality* (LSI) with parameter $C_{\mathsf{LSI}}(\nu)$ if for all smooth functions $f : \mathbb{R}^d \to \mathbb{R}$,

$$\mathsf{Ent}_\nu(f^2) \leq 2C_{\mathsf{LSI}}(\nu) \, \mathbb{E}_\nu[\|\nabla f\|^2] \,, \tag{LSI-I}$$

where $\mathsf{Ent}_\nu(f^2) := \mathbb{E}_\nu[f^2 \log f^2] - \mathbb{E}_\nu[f^2] \log(\mathbb{E}_\nu[f^2])$. Equivalently, for any probability measure $\mu$ over $\mathbb{R}^d$ with $\mu \ll \nu$,

$$2\,\mathsf{KL}(\mu \,\|\, \nu) \leq C_{\mathsf{LSI}}(\nu) \, \mathsf{FI}(\mu \,\|\, \nu) \,, \tag{LSI-II}$$

where $\mathsf{FI}(\mu \,\|\, \nu) := \mathbb{E}_\mu[\|\nabla \log \tfrac{\mathrm{d}\mu}{\mathrm{d}\nu}\|^2]$ is the *Fisher information* of $\mu$ with respect to $\nu$.

## 3 Diffusion for uniform sampling

We propose the following In-and-Out[1] sampler for uniformly sampling from $\mathcal{K}$. Each iteration consists of two steps, one that might leave the body and the second accepted only if it is (back) in $\mathcal{K}$.

It might be illuminating for the reader to compare this algorithm to the well-studied Ball walk (Algorithm 2); each proposed step is a uniform random point in a fixed-radius ball around the current point, and is accepted only if the proposed point is in the body $\mathcal{K}$. In contrast, each iteration of In-and-Out is a two-step process, where the first step (Line 2) ignores the boundary of the body, and

---

[1]This name reflects the "geometry" of how the iterates are moving. As we elaborate in Remark 1, the name 'proximal sampler' may be more familiar to those from an optimization background.

---

**Algorithm 1** In-and-Out

> **Input:** initial point $x_0 \sim \pi_0$, convex body $\mathcal{K} \subset \mathbb{R}^d$, iterations $T$, threshold $N$, and $h > 0$.
> **Output:** $x_{T+1}$.
> 1: **for** $i = 0, \ldots, T$ **do**
> 2:     Sample $y_{i+1} \sim \mathcal{N}(x_i, hI_d)$.
> 3:     Repeat: Sample $x_{i+1} \sim \mathcal{N}(y_{i+1}, hI_d)$ until $x_{i+1} \in \mathcal{K}$ or #attempts$_i \geq N$ (declare **Failure**).
> 4: **end for**

---

the second step (Line 3) is accepted only if a proposal $x_{i+1}$ is a feasible point in $\mathcal{K}$. We will presently elaborate on the benefits of this variation.

Each successful iteration of the algorithm, i.e., one that is not declared "Failure", can be called a *proper* step. We will see that the number of proper steps is directly bounded by isoperimetric constants (such as Poincaré and log-Sobolev) of the target distribution. In fact, this holds quite generally without assuming the convexity of $\mathcal{K}$. The implementation of an iteration is based on rejection sampling (Line 3), and our analysis of the efficiency of this step relies crucially on the convexity of $\mathcal{K}$. This is reminiscent of the Speedy walk in the literature on convex body sampling (Algorithm 3), which is used as a tool to analyze proper steps of the Ball walk. We refer the reader to Appendix C for a brief survey on these and related walks.

This simple algorithm can be interpreted as a composition of "flows" in the space of measures. This view will allow us to use tools from stochastic analysis. In particular, we shall demonstrate how to interpret the two steps of one iteration of In-and-Out as alternating *forward* and *backward* heat flows. We begin by defining an augmented probability measure on $\mathbb{R}^d \times \mathbb{R}^d$ by

$$\pi(x, y) \propto \exp\left(-\tfrac{1}{2h}\|x - y\|^2\right) \mathbb{1}_{\mathcal{K}}(x).$$

We denote by $\pi^X, \pi^{X|Y}(\cdot|y)$ the marginal distribution of its first component (resp. conditional distribution given the second component), and similarly denote by $\pi^Y, \pi^{Y|X}(\cdot|x)$ for the second component. In particular, the marginal in the first component $\pi^X$ is the uniform distribution over $\mathcal{K}$. Sampling from such a joint distribution to obtain the marginal on $X$ (say), can be more efficient than directly working only with $\pi^X$. This idea was utilized in Gaussian Cooling [6] and later as the restricted Gaussian Oracle (RGO) [26, 27].

Under this notation, Algorithm 1 corresponds to a Gibbs sampling scheme from the two marginals of $\pi(x, y)$. To be precise, Line 2 and Line 3 correspond to sampling from

$$y_{i+1} \sim \pi^{Y|X}(\cdot|x_i) = \mathcal{N}(x_i, hI_d) \qquad \text{and} \qquad x_{i+1} \sim \pi^{X|Y}(\cdot|y_{i+1}) = \mathcal{N}(y_{i+1}, hI_d)|_{\mathcal{K}}.$$

We implement the latter step through rejection sampling; if the number of trials in Line 3 hits the threshold $N$, then we halt and declare *failure* of the algorithm. It is well known that such a Gibbs sampling procedure will ensure the desired stationarity of $\pi(x, y)$. Note that, conditioned on the event that the algorithm does not fail, the resulting iterate will be an unbiased sample from the correct distribution.

**Stochastic perspective: forward and backward heat flows.** Our algorithm can be viewed through the lens of stochastic analysis, due to an improved analysis for the proximal sampling [27]. This view provides an interpolation in continuous-time, which is simple and powerful. To make this concrete, we borrow an exposition from [28, §8.3]. We denote the successive laws of $x_i$ and $y_i$ by $\mu_i^X$ and $\mu_i^Y$, respectively. Recall that the first step of sampling from $\pi^{Y|X}(\cdot|x_i)$ (Line 2) yields $\mu_{i+1}^Y = \int \pi^{Y|X=x} \, d\mu_i^X(x)$. This is the result of evolving a probability measure under *(forward) heat flow* of $\mu_i^X$ for some time $h$, given by the following stochastic differential equation: for $Z_0 \sim \mu_i^X$,

$$dZ_t = dB_t \tag{FH}$$

where $B_t$ is the standard Brownian process. We write $\mathsf{law}(Z_t) = \mu_i^X P_t$. In particular, $Z_h = Z_0 + \zeta \sim \mu_i^X * \mathcal{N}(0, hI_d) = \mu_{i+1}^Y$ for $\zeta \sim \mathcal{N}(0, hI_d)$. When $\mu_i^X = \pi^X$, the first step of Algorithm 1 gives

$$\pi^Y(y) = [\pi^X * \mathcal{N}(0, hI_d)](y) = \frac{1}{\mathrm{vol}(\mathcal{K})\,(2\pi h)^{d/2}} \int_{\mathcal{K}} \exp\left(-\frac{1}{2h}\|y - x\|^2\right) dx. \tag{3.1}$$

Table 1: The Fokker-Planck equations for the forward and backward heat flow describe how the laws of $Z_t$ and $Z_t^{\leftarrow}$ in (FH) and (BH) evolve over time. See Appendix B.2 for details.

| | Forward flow | Backward flow |
|---|---|---|
| SDE | $\mathrm{d}Z_t = \mathrm{d}B_t$ | $\mathrm{d}Z_t^{\leftarrow} = \nabla \log(\pi^X P_{h-t})(Z_t^{\leftarrow})\,\mathrm{d}t + \mathrm{d}B_t$ |
| Fokker-Planck | $\partial_t \mu_t = \frac{1}{2}\Delta\mu_t$ | $\partial_t \mu_t^{\leftarrow} = -\operatorname{div}\big(\mu_t^{\leftarrow}\nabla\log(\pi^X P_{h-t})\big) + \frac{1}{2}\Delta\mu_t^{\leftarrow}$ |

The second step of sampling from $\pi^{X|Y}(\cdot|y_{i+1})$ can be represented by $\mu_{i+1}^X = \int \pi^{X|Y=y}\,\mathrm{d}\mu_{i+1}^Y(y)$ (Line 3). The continuous-time process corresponding to this step might not be obvious. However, let us consider (FH) with $Z_0 \sim \pi^X$. Then, $Z_h \sim \pi^Y$, so the joint distribution of $(Z_0, Z_h)$ is simply $\pi$. This implies that $(Z_0|Z_h = y) \sim \pi^{X|Y=y}$. Imagine there is an SDE *reversing* the forward heat flow in a sense that if we are initialized deterministically at $Z_h \sim \delta_y$ at time 0, then the law of the SDE at time $h$ would be $\mathsf{law}(Z_0|Z_h = y) = \pi^{X|Y=y}$. Then, this SDE would serve as a continuous-time interpolation of the second step.

Such a *time reversal* SDE is indeed possible! The following SDE on $(Z_t^{\leftarrow})_{t\in[0,h]}$ initialized at $Z_0^{\leftarrow} \sim \pi^Y = \pi^X P_h$ ensures $Z_{h-t} \sim \mathsf{law}(Z_t^{\leftarrow}) = \pi^X P_{h-t}$:

$$\mathrm{d}Z_t^{\leftarrow} = \nabla\log(\pi^X P_{h-t})(Z_t^{\leftarrow})\,\mathrm{d}t + \mathrm{d}B_t \quad \text{for } t\in[0,h]. \tag{BH}$$

Although this is designed to reverse (FH) **initialized by** $Z_0 \sim \pi^X$ (so $Z_h = Z_0^{\leftarrow} \sim \pi^Y$), its construction also ensures that if $Z_0^{\leftarrow} \sim \delta_y$, a point mass, then $Z_h^{\leftarrow} \sim \mathsf{law}(Z_0|Z_h = y) = \pi^{X|Y=y}$. Thus, if we initialize (BH) with $Z_0^{\leftarrow} \sim \mu_{i+1}^Y$, then the law of $Z_h^{\leftarrow}$ corresponds to $\int \pi^{X|Y=y}\,\mathrm{d}\mu_{i+1}^Y(y) = \mu_{i+1}^X$.

*Remark* 1. We note that In-and-Out is exactly the *proximal* sampling scheme [26, 27, 29] for uniform distributions. The proximal sampler with a target density proportional to $\exp(-V(x))$ considers an augmented distribution $\pi(x,y) \propto \exp(-V(x) - \frac{1}{2h}\|x-y\|^2)$ and then repeats the following two steps: (1) $y_{i+1} \sim \pi^{Y|X=x_i} = \mathcal{N}(x_i, hI_d)$ and then (2) $x_{i+1} \sim \pi^{X|Y=y_{i+1}}$. Naïvely, the proximal sampler is implemented by performing rejection sampling, with the Gaussian centered at the minimizer of $\log\pi^{\cdot|Y=y_{i+1}}$ as the proposal. Realizing this would require a projection oracle (to $\mathcal{K}$), which is only known to be implementable with $\mathcal{O}(d^2)$ membership queries. In-and-Out completely avoids the need for a projection oracle.

## 4 Results

Our computational model is the classical general model for convex bodies [30]. We assume $\mathrm{vol}(\mathcal{K}) > 0$ throughout this paper. Below, $B_r(x)$ denotes the $d$-dimensional ball of radius $r$ centered at $x$.

**Definition 4** (Convex body oracle). A *well-defined membership oracle* for a convex body $\mathcal{K} \subset \mathbb{R}^d$ is given by a point $x_0 \in \mathcal{K}$, a number $D > 0$, with the guarantee that $B_1(x_0) \subseteq \mathcal{K} \subseteq B_D(x_0)$, and an oracle that correctly answers *YES* or *NO* to any query of the form "$x \in \mathcal{K}$?"

**Definition 5** (Warmness). A distribution $\mu$ is $M$-*warm* with respect to another distribution $\pi$ if for every $x$ in the support of $\pi$, we have $\mathrm{d}\mu(x) \leq M\,\mathrm{d}\pi(x)$.

We now summarize our main result, which is further elaborated in Appendix B.4 (Theorem 5). Below, $\pi^{\mathcal{K}}$ is the uniform distribution over $\mathcal{K}$, and $\mathcal{R}_q$ is the Rényi-divergence of order $q$ (see Definition 1).

**Theorem 1** (Informal version of Theorem 5)**.** *For any given $\eta, \varepsilon \in (0,1)$, $q \geq 1$, and any convex body $\mathcal{K}$ given by a well-defined membership oracle, there exist choices of parameters $h, N$ such that* In-and-Out*, starting from an $M$-warm distribution, with probability at least $1 - \eta$, returns $X \sim \mu$ such that $\mathcal{R}_q(\mu \| \pi^{\mathcal{K}}) \leq \varepsilon$. The number of proper steps is $\widetilde{\mathcal{O}}(qd^2\Lambda\log^2\frac{M}{\eta\varepsilon})$, and the expected total number of membership queries is $\widetilde{\mathcal{O}}(qMd^2\Lambda\log^6\frac{1}{\eta\varepsilon})$, where $\Lambda$ is the largest eigenvalue of the covariance of $\pi^{\mathcal{K}}$.*

*Remark* 2. Despite our guarantee being in the much stronger "metric" of $\mathcal{R}_q$ compared to the TV guarantees of Ball walk, we do not have to incur any additional asymptotic complexity.

To obtain this result, one should choose the following values for the parameters: $h^{-1} = \widetilde{\Theta}(d^2\log\frac{qM\Lambda}{\eta}\log\log\frac{1}{\varepsilon})$, $N = \widetilde{\Theta}(\frac{qMd^2\Lambda\log^5(1/\varepsilon)}{\eta})$. See Lemma 3 for more details.

Finally, while the assumption of warmness for the initialization may seem strong at the outset, for well-rounded convex bodies ($\mathbb{E}_{X \sim \pi}[\|X\|^2] \leq C^2 d$ for some constant $C$), it is possible to generate an $\mathcal{O}(1)$ warm-start with complexity $\widetilde{\mathcal{O}}(d^3)$. See [6, 31] for details.

We note that for $X \sim \pi^{\mathcal{K}}$,

$$\|\mathrm{Cov}(\pi^{\mathcal{K}})\|_{\mathsf{op}} \leq \mathrm{tr}(\mathrm{Cov}(\pi^{\mathcal{K}})) = \mathbb{E}[\|X - \mathbb{E}X\|^2] \leq D^2\,.$$

The above guarantee in the Rényi divergence immediately provides $\mathcal{W}_2$, TV, KL, and $\chi^2$ guarantees as special cases. For two distributions $\mu$ and $\pi$, we have

1. $\mathsf{KL}(\mu\|\pi) = \lim_{q \downarrow 1} \mathcal{R}_q(\mu\|\pi) \leq \mathcal{R}_q(\mu\|\pi) \leq \mathcal{R}_{q'}(\mu\|\pi) \leq \mathcal{R}_\infty(\mu\|\pi) = \log \sup \frac{\mathrm{d}\mu}{\mathrm{d}\pi}, 1 < q \leq q'$.
2. $2\|\mu - \pi\|_{\mathsf{TV}}^2 \leq \mathsf{KL}(\mu \,\|\, \pi) \leq \log(\chi^2(\mu \,\|\, \pi) + 1) = \mathcal{R}_2(\mu \,\|\, \pi)$.
3. $\mathcal{W}_2^2(\mu, \pi) \leq 2C_{\mathsf{LSI}}(\pi)\,\mathsf{KL}(\mu \,\|\, \pi)$ (Talagrand's $\mathsf{T}_2$-inequality) and $C_{\mathsf{LSI}}(\pi^{\mathcal{K}}) \lesssim D^2$.
4. $\mathcal{W}_2^2(\mu, \pi) \leq 2C_{\mathsf{PI}}(\pi)\,\chi^2(\mu \,\|\, \pi)$ [32] and $C_{\mathsf{PI}}(\pi^{\mathcal{K}}) \lesssim \|\mathrm{Cov}(\pi^{\mathcal{K}})\|_{\mathsf{op}} \log d$.

The query complexity is better if the convex body is (near-)*isotropic*, i.e., the uniform distribution over the body has (near-)identity covariance. This relies on recent estimates of the worst-case Poincaré constant for isotropic log-concave distributions [33, 34]. The condition that the convex body is isotropic can be achieved in practice through a *rounding* procedure [35]. See §5 for more details.

**Corollary 1.** *Assume that $\pi^{\mathcal{K}}$ is near-isotropic, i.e. the operator norm of its covariance is $\mathcal{O}(1)$. Under the same setting as above,* In-and-Out *succeeds with probability $1 - \eta$, returning $X \sim \mu$ such that $\mathcal{R}_q(\mu \,\|\, \pi^{\mathcal{K}}) \leq \varepsilon$. The number of proper steps is $\widetilde{\mathcal{O}}(qd^2 \log^2 \frac{M}{\eta\varepsilon})$, and the expected total number of membership queries is $\widetilde{\mathcal{O}}(qMd^2 \log^6 \frac{1}{\eta\varepsilon})$.*

Our analysis will in fact show that the bound on the number of proper steps holds for general *non-convex bodies* and *any feasible start* in $\mathcal{K}$. This is deduced under an $M$-warm start in Corollaries 2 and 3. We remark that such a bound for non-convex uniform sampling is not known for the Ball walk or the Speedy walk.

**Theorem 2.** *For any given $\varepsilon \in (0, 1)$ and set $\mathcal{K} \subset B_D(0)$ with $\mathrm{vol}(\mathcal{K}) > 0$,* In-and-Out *with variance $h$ and $M$-warm initial distribution achieves $\mathcal{R}_q(\mu_m^X \,\|\, \pi^X) \leq \varepsilon$ after the following number of iterations:*

$$m = \min \begin{cases} \mathcal{O}\big(qh^{-1}C_{\mathsf{PI}}(\pi^X) \log \frac{M}{\varepsilon}\big) & \text{for } q \geq 2\,, \\ \mathcal{O}\big(qh^{-1}C_{\mathsf{LSI}}(\pi^X) \log \frac{\log M}{\varepsilon}\big) & \text{for } q \geq 1\,. \end{cases}$$

We have two different convergence results above under (LSI-I) and (PI). Under (LSI-I) we have a *doubly-logarithmic* dependence on the warmness parameter $M$. On the other hand, using (PI), which is weaker than (LSI-I) (in general, $C_{\mathsf{PI}} \leq C_{\mathsf{LSI}}$), the dependence on $M$ is logarithmic. We discuss implications of our results further in §4.2.

## 4.1 Outline of analysis.

We first record two fundamental lemmas, which introduce the mathematical formalism for our analysis. The first is the existence of forward and backward heat flows (Lemma 12), which will interpolate each line in Algorithm 1. These flow equations describe how the laws of $Z_t$ and $Z_t^{\leftarrow}$ in (FH) and (BH) evolve respectively over time. All proofs are deferred to Appendix B.

**Lemma 1.** *The forward heat flow equation with initial distribution $\mu_0$ is given by*

$$\partial_t \mu_t = \tfrac{1}{2}\Delta\mu_t\,,$$

*and its backward heat flow equation is given by*

$$\partial_t \mu_t^{\leftarrow} = -\,\mathrm{div}\big(\mu_t^{\leftarrow}\nabla\log(\pi^X P_{h-t})\big) + \tfrac{1}{2}\Delta\mu_t^{\leftarrow} \quad \text{with } \mu_0^{\leftarrow} = \mu_h\,.$$

*These admit (weak) solutions on $[0, h]$ for any initial distribution $\mu_0$ with $\frac{\mathrm{d}\mu_0}{\mathrm{d}\pi^X} \leq M < \infty$.*

One successful iteration of In-and-Out is exactly the same as the composition of running the forward heat flow and then backward heat flow, both for time $h$.

**Lemma 2.** *Let $\mu_k^X$ be the law of the $k$-th iterate $x_k$ of* In-and-Out. *If* (FH) *is initialized with* $\mathsf{law}(Z_0) = \mu_k^X$, *then* $\mathsf{law}(Z_h) = \mu_{k+1}^Y$. *If* (BH) *is initialized with* $\mathsf{law}(Z_0^\leftarrow) = \mu_{k+1}^Y$, *then* $\mathsf{law}(Z_h^\leftarrow) = \mu_{k+1}^X$.

We summarize our proof strategy below, which requires us to demonstrate two facts: (i) The current distribution should converge to the uniform distribution, (ii) within each iteration of the algorithm, the failure probability and the expected number of rejections should be small enough. In this section we provide the main claims within each of these parts, and defer the remaining details to Appendix B.

While each individual component resembles pre-existing work in the literature, in their synthesis we will demonstrate how to interleave past developments in theoretical computer science, optimal transport, and functional analysis. The combination of these in this domain yields elegant and surprisingly simple proofs, as well as stronger results.

**Part (i).** Broadly speaking, we need to demonstrate that the corresponding Markov chain is rapidly mixing. Here, we use the heat flow perspective to derive mixing rates under any suitable divergence measure (such as KL, $\chi^2$, or $\mathcal{R}_q$). This extends known results for the unconstrained setting [27]. To summarize the proof, by considering instead the solutions after small time $t$, we invoke known contraction results from [27] and then use a continuity argument to conclude the proof.

**Theorem 3.** *Let $\mu_k^X$ be the law of the $k$-th output of* In-and-Out *with initial distribution $\mu_0^X$. Let $C_{\mathsf{LSI}}$ be the* (LSI-I) *constant of the uniform distribution $\pi^X$ over $\mathcal{K}$. Then, for any $q \geq 1$,*

$$\mathcal{R}_q(\mu_k^X \parallel \pi^X) \leq \frac{\mathcal{R}_q(\mu_0^X \parallel \pi^X)}{(1 + h/C_{\mathsf{LSI}})^{2k/q}}.$$

*For $C_{\mathsf{PI}}$ the* (PI) *constant of $\pi^X$,*

$$\chi^2(\mu_k^X \parallel \pi^X) \leq \frac{\chi^2(\mu_0^X \parallel \pi^X)}{(1 + h/C_{\mathsf{PI}})^{2k}}.$$

*Furthermore, for any $q \geq 2$,*

$$\mathcal{R}_q(\mu_k^X \parallel \pi^X) \leq \begin{cases} \mathcal{R}_q(\mu_0^X \parallel \pi^X) - \frac{2k \log(1 + h/C_{\mathsf{PI}})}{q} & \text{if } k \leq \frac{q(\mathcal{R}_q(\mu_0^X \parallel \pi^X) - 1)}{2\log(1 + h/C_{\mathsf{PI}})}, \\ (1 + h/C_{\mathsf{PI}})^{-2(k-k_0)/q} & \text{if } k \geq k_0 := \left\lceil \frac{q(\mathcal{R}_q(\mu_0^X \parallel \pi^X) - 1)}{2\log(1 + h/C_{\mathsf{PI}})} \right\rceil. \end{cases}$$

The final result reduces the problem of obtaining a mixing guarantee to that of demonstrating a functional inequality on the target distribution. For this, it is not strictly necessary that $\mathcal{K}$ be convex.

**Part (ii).** Convexity of $\mathcal{K}$ is crucial this time unlike Part (i). We show in Appendix B.3 that the failure probability remains under control by taking a suitable variance $h$ and threshold $N$, and that the expected number of trials per iteration is of order $\log N$, not $N$. To do this, we apply a detailed argument involving local conductance and the convexity of $\mathcal{K}$, which relies on techniques from [36].

**Lemma 3** (Per-iteration guarantees). *Let $\mathcal{K}$ be any convex body in $\mathbb{R}^d$ presented by a well-defined membership oracle, $\pi^X$ the uniform distribution over $\mathcal{K}$, and $\mu$ an $M$-warm initial distribution with respect to $\pi^X$. For any given $m \in \mathbb{N}$ and $\eta \in (0, 1)$, set $Z = \frac{9mM}{\eta} (\geq 9)$, $h = \frac{\log \log Z}{2d^2 \log Z}$ and $N = Z \log^4 Z = \widetilde{\mathcal{O}}(\frac{mM}{\eta})$. Then, the failure probability of one iteration of* In-and-Out *is at most $\eta/m$, and the expected membership queries per iteration is $\mathcal{O}(M \log^4 \frac{mM}{\eta})$.*

## 4.2 Discussion

**No need to be lazy.** Previous uniform samplers like Ball walk are made *lazy* (i.e., with probability $1/2$, it does nothing), to ensure convergence to the target stationary distribution. However, our algorithm does not need this, as our sampler is shown to directly contract towards the target.

**Unified framework.** We remark that Theorem 2 places the previously known mixing guarantees for Ball walk, Speedy walk in a unified framework. Existing tight guarantees for Speedy walk are in TV distance and based on the log-Sobolev constant, assuming an oracle for implementing each step [37].

The known convergence guarantees of Ball walk (see Appendix C for details), namely the mixing time of $\widetilde{\mathcal{O}}(Md^2D^2 \log \frac{1}{\varepsilon})$ for TV distance, are for the composite algorithm [Speedy walk+rejection sampling]. Here Speedy walk records only the accepted steps of Ball walk, so its stationary distribution differs slightly from the uniform distribution (and can be corrected with a post-processing step). On the other hand, In-and-Out actually converges to $\pi^{\mathcal{K}}$ without any adjustments and achieves stronger Rényi divergence bounds in the same asymptotic complexity. Our analysis shows that the mixing guarantee is determined by isoperimetric constants of the target (Poincaré or log-Sobolev).

**Effective step size.** The Ball walk's largest possible step size is of order $1/\sqrt{d}$ (see Appendix C) to keep the rejection probability bounded by a constant. This bound could also be viewed as an "effective" step size of In-and-Out, since the $\ell_2$-norm of the Gaussian $\mathcal{N}(0, hI)$ is concentrated around $\sqrt{hd}$ and we will set the variance $h$ of In-and-Out to $\widetilde{\mathcal{O}}(1/d^2)$, so we have $\sqrt{hd} \approx 1/\sqrt{d}$.

**What has really changed?** In-and-Out has clear similarities to both Ball walk and Speedy walk. What then are the changes that allow us to use continuous-time interpolation? One step of Ball walk is [random step $(y \in B_\delta(x))$ + Metropolis-filter (accept if $y \in \mathcal{K}$)]. This filtering is an abrupt discrete step, and it is unclear how to control contraction. It could be replaced by a step of Speedy walk $(x \sim \mathrm{Unif}(B_\delta(y) \cap \mathcal{K}))$. Then, each iteration of In-and-Out can be viewed as a Gaussian version of a Ball walk's proposal + Speedy walk algorithm.

How can we compare In-and-Out with Speedy walk? Iterating speedy steps leads to a biased distribution. As clarified in Remark 3, one step of (a Gaussian version of) Speedy walk can be understood as a step of backward heat flow. Therefore, if one can control the isoperimetric constants of the biased distribution along the trajectory of the backward flow, then contraction of Speedy walk toward the biased distribution will follow from the simultaneous backward analysis.

## 5 Related work

Sampling from constrained log-concave distributions is a fundamental task arising in many fields. Uniform sampling with convex constraints is its simplest manifestation, which was first studied as a core subroutine for a randomized volume-computation algorithm [1]. Since then, this fundamental problem has been studied for over three decades [2, 3, 4, 38, 5, 25, 24]. We review these algorithms, grouping them under three categories — geometric random walks, structured samplers, and diffusion-type samplers. Below, $\mathcal{K}$ is convex.

**Geometric random walk.** We discuss two geometric random walks – Ball walk [3, 4] and Hit-and-Run [39, 17]. Ball walk is a simple metropolized random walk; it draws $y$ uniformly at random from a ball of radius $\delta$ centered at a current point $x$, and moves to $y$ if $y \in \mathcal{K}$ and stays at $x$ otherwise. In the literature, Ball walk actually refers to a composite algorithm consisting of [Speedy walk+ rejection sampling], where Speedy walk records only the accepted steps of Ball walk (see Appendix C for details). The step size $\delta$ should be set to $\mathcal{O}(d^{-1/2})$ to avoid stepping outside of $\mathcal{K}$. [4] showed that Ball walk needs $\widetilde{\mathcal{O}}(Md^2D^2 \log \frac{1}{\varepsilon})$ membership queries to be $\varepsilon$-close to $\pi^{\mathcal{K}}$ in TV, where $D$ is the diameter of $\mathcal{K}$, and the warmness parameter $M$ measures the closeness of the initial distribution to the target uniform distribution $\pi^{\mathcal{K}}$.

Hit-and-Run is another zeroth-order algorithm that needs *no step size*; it picks a uniform random line $\ell$ passing a current point, and move to a uniform random point on $\ell \cap \mathcal{K}$. [5] shows that, if we define the second moment as $R^2 := \mathbb{E}_{X \sim \pi^{\mathcal{K}}}[\|X - \mathbb{E}X\|^2]$, then Hit-and-Run requires $\mathcal{O}(d^2R^2 \log \frac{M}{\varepsilon})$ queries. Notably, this algorithm has a poly-logarithmic dependence on $M$ as opposed to Ball walk.

Both algorithm are affected by skewed shape of $\mathcal{K}$ (i.e., large $D$ or $R$), so these samplers are combined with pre-processing step called *rounding*. This procedure finds a linear transformation that makes the geometry of $\mathcal{K}$ less skewed and so more amenable to sampling. In literature, there exists a randomized algorithm [35] that rounds $\mathcal{K}$ and generates a good warm start (i.e., $M = \mathcal{O}(1)$), with Ball walk used as a core subroutine. This algorithm takes up $\widetilde{\mathcal{O}}(d^{3.5})$ queries in total, and in such position with the good warm start, Ball walk only needs $\widetilde{\mathcal{O}}(d^2 \log \frac{1}{\varepsilon})$ queries to sample from $\pi^{\mathcal{K}}$.

**Structured samplers.** The aforementioned samplers based on geometric random walks require *only* access to the membership oracle of the convex body *without* any additional structural assumptions.

The alternate paradigm of *geometry-aware sampling* attempts to exploit the *structure* of convex constraints, with the aim of expediting the convergence of the resultant sampling schemes. One common assumption is to make available a *self-concordant barrier function* $\phi$ which has regularity on its high-order derivatives and blows up when approaching the boundary $\partial \mathcal{K}$. The Hessian of $\phi$ encodes the local geometry of the constraint, and the samplers often work directly with $\nabla^2 \phi$.

The first canonical example of such a zeroth-order sampler is Dikin walk used when $\mathcal{K}$ is given by $m$ linear constraints [40]; it draws a uniform sample from an ellipsoid (characterized by $\nabla^2 \phi$) of fixed radius around a current point, and is often combined with a Metropolis adjustment. [40] shows that Dikin walk mixes in $\mathcal{O}(md \log \frac{M}{\varepsilon})$ steps, although each iteration is slightly more expensive than one membership query. This algorithm requires no rounding, but still needs a good warm-start, which can be achieved by an annealing-type algorithm using $\widetilde{\mathcal{O}}(md)$ iterations of Dikin walk [41].

Riemannian Hamiltonian Monte Carlo is a structured sampler that exploits the first-order information of the potential (i.e., $\nabla \log(1/\pi)$) [42]; its proposal is given as the solution to the Hamilton's ODE equation, followed by the Metropolis-filter. In the linear-constraint setting above, it requires $\mathcal{O}(md^{2/3} \log \frac{M}{\varepsilon})$ many iterations to achieve $\varepsilon$-close distance to $\pi^{\mathcal{K}}$ [18]. This sampler is further analyzed for practical ODE solvers [43] and for more sophisticated self-concordant barriers [44].

Similarly, Mirror Langevin [45, 46, 47, 48] is a class of algorithms which converts the constrained problem into an unconstrained one obtained by considering the pushforward of the constrained space by $\nabla \phi$. The algorithm can also be metropolized [49]. The best known rate for this algorithm is $\widetilde{\mathcal{O}}(d \log \frac{1}{\varepsilon})$ under some strong assumptions on $\phi$.

**Diffusion-type samplers.** Samplers based on discretizations of Itô diffusions, stochastic processes which rapidly mix to $\pi$ in continuous time, have long been used for sampling without constraints [19, 20, 21, 28]. While the underlying stochastic processes generalize easily to constrained settings, the discretization analysis relies crucially on the smoothness of the target distribution. This is clearly impossible to achieve in the constrained setting, so some techniques are required to circumvent this difficulty. These algorithms, however, generalize easily to the more general problem of sampling from distributions of the form $\tilde{\pi}^X \propto e^{-f} \mathbb{1}_{\mathcal{K}}$, by incorporating first order information from $f$.

The first approach for adapting diffusion-based samplers [50, 25, 51] iterates a two-step procedure. First, a random step is taken, with $x_{k+1/2} \sim \mathcal{N}(x_k, 2hI_d)$ for some appropriately chosen step $h$,[2] and then project it to $\mathcal{K}$, i.e., $x_{k+1} = \mathsf{proj}_{\mathcal{K}}(x_{k+1/2})$. The complexity is given in terms of queries to a *projection oracle*, each call to which can be implemented with a polynomial number of membership oracle queries; a total of $\tilde{\mathcal{O}}(d^2 D^3/\varepsilon^4)$ queries are needed to be $\varepsilon$-close in $\mathcal{W}_2$ to $\pi^X$. Another approach, which uses an algorithmically designed "soft" penalty instead of a projection, was proposed in [52], and achieves a rate estimate of $\tilde{\mathcal{O}}(d/\varepsilon^{10})$.

A second approach, suggested by [24], considers a different proximal scheme, which performs a "soft projection" onto $\mathcal{K}$, by taking steps like $\mathcal{N}((1 - h\lambda^{-1})x_k + h\mathsf{proj}_{\mathcal{K}}(x_k), 2hI_d)$. It is called Moreau-Yosida regularized Langevin, named after an analogous regularization scheme for constrained optimization. This scheme also relies on access to a projection oracle for $\mathcal{K}$, and quantifies their query complexity accordingly. Their final rate estimate is $\tilde{\mathcal{O}}(d^5/\varepsilon^6)$ to be $\varepsilon$-close in TV distance to $\pi^X$.

Observing the prior work integrating diffusion-based sampling with convex constraints, the dependence on the key parameters $d, \varepsilon$, while polynomial, are many orders worse than the rates for zeroth-order samplers such as Ball walk, Hit-and-Run. In contrast, our analysis not only recovers but in some sense surpasses the known rates for Ball walk, Hit-and-Run, while harmonizing well with the continuous-time perspective of diffusions.

**Proximal schemes for sampling.** The Gibbs sampling scheme used in this paper was inspired by the restricted Gaussian oracle introduced in [26] (in turn inspired by Gaussian Cooling [6]), which alternately iterates between a pure Gaussian step, and a "proximal" step (which we elaborate in our exposition). This scheme was given novel interpretations by [27], which showed that it interpolates the forward and backward heat flows, in the sense defined by [53]. The backward heat flow itself is intimately related to stochastic localization schemes, invented and popularized in [54, 55].

---

[2]A gradient step can be added in the more general case, for sampling from $\tilde{\pi}^X$.

This formulation proved surprisingly powerful, allowing many existing rates in unconstrained sampling to be recovered from a relatively simple analysis. This was further extended by [29] to achieve the current state-of-the-art rate in unconstrained sampling. Finally, [56] suggest that this could be applied to tackle some constrained problems. However, the assumptions in this final mentioned work are not compatible with the uniform sampling problem on general convex bodies.

## 6 Conclusion

We propose In-and-Out for uniform sampling on convex bodies, and show that it obtains guarantees in $\mathcal{R}_q$ divergence from an $M$-warm start, substantially stronger than prior work, and without increasing the computational complexity. Notably, our proof technique is quite different and provides a direct reduction to isoperimetric constants of the target distribution.

While the current work focuses on uniform sampling of convex bodies, there are a number of natural extensions that may be considered. It may be possible to prove analogous results for general log-concave distributions, and on non-log-concave distributions satisfying isoperimetric inequalities, e.g., it is open to find a polytime algorithm for sampling a general distribution satisfying a Poincaré inequality presented by a function oracle (with no smoothness assumptions).

**Acknowledgements.** We are deeply grateful to Andre Wibisono and Sinho Chewi for helpful comments and pointers to the literature for Lemma 10. This work was supported in part by NSF award 2106444, NSERC through the CGS-D award, and a Simons Investigator award.

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

# A   Helpful lemmas

Before proceeding, we state two important lemmas which are needed for our proofs. The first is the data-processing inequality for Rényi divergence and $f$-divergence, given below.

**Lemma 4** (Data-processing inequality). *For measures $\mu, \nu$, Markov kernel $P$, $f$-divergence $D_f$, and $q \geq 1$, it holds that*

$$D_f(\mu P \parallel \nu P) \leq D_f(\mu \parallel \nu), \quad \text{and} \quad \mathcal{R}_q(\mu P \parallel \nu P) \leq \mathcal{R}_q(\mu \parallel \nu).$$

Functional inequalities allow us to show exponential contraction of various divergences, through the following helpful inequality.

**Lemma 5** (Grönwall). *Suppose that $u, g : [0, T] \to \mathbb{R}$ are two continuous functions, with $u$ being differentiable on $[0, T]$ and satisfying*

$$u'(t) \leq g(t)\, u(t) \qquad \text{for all } t \in [0, T].$$

*Then,*

$$u(t) \leq \exp\!\Big(\int_0^t g(s)\,\mathrm{d}s\Big)\, u(0) \qquad \text{for all } t \in [0, T].$$

# B   Analysis

Before proceeding, we first give the proof of Lemma 2 in the main text. A more general version of Lemma 1 is proved in the form of Lemma 12.

*Proof of Lemma 2.* We explicitly show that the forward and backward heat flows indeed interpolate the two discrete steps given in Algorithm 1. For the forward part, we have $Z_h = Z_0 + \zeta$ for $\zeta \sim \mathcal{N}(0, hI_d)$, so

$$\mathsf{law}(Z_h) = \mathsf{law}(Z_0) * \mathcal{N}(0, hI_d) = \mu_k^X * \mathcal{N}(0, hI_d) = \mu_{k+1}^Y.$$

Regarding the backward part, it is known from [27, Lemma 14] that the construction of the time-reversal SDE ensures that $(Z_h^\leftarrow, Z_0^\leftarrow)$ and $(Z_0, Z_h)$ have the same joint distribution, when $Z_0 \sim \pi^X$ (and so $Z_h \sim \pi^Y$). Hence, $\mathsf{law}(Z_h^\leftarrow | Z_0^\leftarrow = y) = \mathsf{law}(Z_0 | Z_h = y) = \pi^{X|Y=y}$, where the last equality follows from $(Z_0, Z_h) \sim \pi$. Since we initialize (BH) with $Z_0^\leftarrow = y \sim \mu_{k+1}^Y$, we have

$$\mathsf{law}(Z_h^\leftarrow) = \int \mathsf{law}(Z_h^\leftarrow | Z_0^\leftarrow = y)\, \mu_{k+1}^Y(\mathrm{d}y) = \int \pi^{X|Y}(\cdot|y)\, \mu_{k+1}^Y(\mathrm{d}y) = \mu_{k+1}^X,$$

where the last follows from the definition of Line 3. $\qquad\square$

## B.1   Functional inequalities

The contraction of an outer loop of our algorithm is controlled by isoperimetry of the uniform distribution $\pi^X$, which is described precisely by a functional inequality. The most natural ones to consider in this setting are the Poincaré inequality (PI) and log-Sobolev inequality (LSI-I). In Appendix D, we provide a more detailed discussion of how these are related to other important notions of isoperimetry, such as the *Cheeger* and *log-Cheeger* inequalities.

Below, we use $\mu, \nu$ to denote two arbitrary probability measures over $\mathbb{R}^d$. The relationship between a Poincaré inequality and the $\chi^2$-divergence is derived by substituting $f = \frac{\mathrm{d}\nu}{\mathrm{d}\mu}$ into (PI).

**Lemma 6.** *Assume that $\nu$ satisfies (PI) with parameter $C_{\mathsf{PI}}(\nu)$. For any probability measure $\mu$ over $\mathbb{R}^d$ with $\mu \ll \nu$, it holds that*

$$\chi^2(\mu \parallel \nu) \leq \frac{C_{\mathsf{PI}}(\nu)}{2}\, \mathbb{E}_\nu\big[\big\|\nabla \frac{\mathrm{d}\mu}{\mathrm{d}\nu}\big\|^2\big].$$

The Poincaré inequality implies functional inequalities for the Rényi divergence.

**Lemma 7** ([23, Lemma 9]). *Assume that $\nu$ satisfies* (PI) *with parameter $C_{\mathsf{PI}}(\nu)$. For any $q \geq 2$ and probability measure $\mu$ over $\mathbb{R}^d$, it holds that*

$$1 - \exp(-\mathcal{R}_q(\mu \,\|\, \nu)) \leq \frac{q\, C_{\mathsf{PI}}(\nu)}{4}\, \mathsf{RF}_q(\mu \,\|\, \nu)\,,$$

*where $\mathsf{RF}_q(\mu\,\|\,\nu) := q\, \mathbb{E}_\nu\big[\big(\frac{\mathrm{d}\mu}{\mathrm{d}\nu}\big)^q \|\nabla \log \frac{\mathrm{d}\mu}{\mathrm{d}\nu}\|^2\big] \big/ \mathbb{E}_\nu\big[\big(\frac{\mathrm{d}\mu}{\mathrm{d}\nu}\big)^q\big]$ is the* Rényi Fisher information *of order $q$ of $\mu$ with respect to $\nu$.*

The log-Sobolev inequality paired with the KL-divergence (LSI-II) can be understood as a special case of the following inequality[3] paired with the $q$-Rényi divergence for $q \geq 1$.

**Lemma 8** ([23, Lemma 5]). *Assume that $\nu$ satisfies* (LSI-II) *with parameter $C_{\mathsf{LSI}}(\nu)$. For any $q \geq 1$ and probability measure $\mu$ over $\mathbb{R}^d$, it holds that*

$$\mathcal{R}_q(\mu \,\|\, \nu) \leq \frac{q\, C_{\mathsf{LSI}}(\nu)}{2}\, \mathsf{RF}_q(\mu \,\|\, \nu)\,.$$

*Note that $\lim_{q \to 1} \mathcal{R}_q = \mathsf{KL}$ and $\mathsf{RF}_1 = \mathsf{FI}$.*

We have collected below the functional inequalities used to establish the mixing of our algorithm (see Appendix D for a detailed presentation).

**Lemma 9.** *Let $\mathcal{K} \subset \mathbb{R}^d$ be a convex body with diameter $D$, and $\pi$ be the uniform distribution over $\mathcal{K}$. Then, $C_{\mathsf{PI}}(\pi) \lesssim \|\mathrm{Cov}(\pi)\|_{\mathsf{op}} \log d$ and $C_{\mathsf{LSI}}(\pi) \lesssim D^2$. If $\pi$ is isotropic, then $C_{\mathsf{PI}}(\pi) \lesssim \log d$ and $C_{\mathsf{LSI}}(\pi) \lesssim D$.*

## B.2 Contraction and mixing

We start by analyzing how many outer iterations of In-and-Out are required to be $\varepsilon$-close to $\pi^X$, the uniform distribution over $\mathcal{K}$. The contraction of Algorithm 1 comes from analyzing Lines 2 and 3 through the perspective of heat flows (see §3). To exploit this view, we first revisit the previous contraction analysis in [27], which is carried out for distributions with *smooth* densities. Although the uniform distribution is not even continuous, we prove a technical lemma (Lemma 12) that enables us to extend previously known results to the uniform distribution. Lastly, combining the previous results with our technical lemma, we obtain clean contraction results of Algorithm 1 toward the uniform distribution $\pi^X$ in Theorem 3.

**Part I: Contraction analysis for smooth distributions.** In this part, we review the contraction results for heat flow and its time-reversal [27], which are intimately connected with our algorithm. We also provide key technical ingredients needed for its proof, such as the computations for measures evolving under simultaneous forward/backward heat flows. We refer interested readers to Appendix E for additional details. Only in **Part I**, we assume that $\nu$ denotes a probability measure with smooth density.

**Forward heat flow.** We begin by introducing the "heat flow" equation (or also known as the *Fokker-Planck* equation), which describes the evolution of the law of $Z_t$ under (FH),

$$\partial_t \mu_t = \frac{1}{2}\, \Delta \mu_t = \frac{1}{2}\, \mathrm{div}(\mu_t \nabla \log \mu_t)\,. \tag{FP-FH}$$

It is well known that one can realize this equation in discrete time through a Gaussian transition density, in the sense that, for $\mu_h$ (the solution at time $h > 0$ to (FP-FH) with initial condition $\mu_0$), and for any smooth function $f : \mathbb{R}^d \to \mathbb{R}$,

$$\mathbb{E}_{\mu_h}[f(x)] = \mathbb{E}_{\mu_0}[P_h f(x)]\,,$$

where $P_h f(x) = \mathbb{E}_{\mathcal{N}(x, hI_d)}[f]$.[4] By this we can formally identify $\mu_h = \mu_0 P_h$, and also write $\mu_h$ for the law of $Z_h$, where $\{Z_h\}_{h \geq 0}$ solves (FH).

---

[3]Such inequalities are often called Polyak-Łojasiewicz inequalities, which say for $f : \mathbb{R}^d \to \mathbb{R}$, and all $y \in \mathbb{R}^d$ that $f(y) \leq c\,\|\nabla f(y)\|^2$ for some constant $c$, if $\min f(x) = 0$.

[4]$\{P_h\}_{h \geq 0}$ is often called the heat semigroup.

**Backward heat flow.** Although there are many ways to define a "reversal" of $P_h$, we will use the notion of *adjoint* introduced by [53], which is the most immediately useful.

Given some initial measure $\nu$ and some time horizon $h$, the adjoint corresponds to reversing (FH) for times in $[0, h]$ when the initial distribution under consideration is $Z_0 \sim \nu$. For other measures, it must be interpreted more carefully, and is given by the following partial differential equation starting from some measure $\mu_0^{\leftarrow}$ (see (E.1) and its derivation):

$$\partial_t \mu_t^{\leftarrow} = -\operatorname{div}\big(\mu_t^{\leftarrow} \nabla \log(\nu P_{h-t})\big) + \frac{1}{2}\Delta \mu_t^{\leftarrow} \quad \text{for } t \in [0, h]. \tag{FP-BH}$$

Write $\mu_t^{\leftarrow} = \mu_0^{\leftarrow} Q_t^{\nu,h}$, where $\{Q_t^{\nu,h}\}_{t \in [0,h]}$ is a family of transition densities. Write $\mathbf{P}_{0,h}$ for the joint distribution of the $(Z_0, Z_h)$-marginals of (FH), when $Z_0 \sim \nu$, and $\mathbf{P}_{0|h}$ for the conditional. Note that $\mathbf{P}_{h|0}(\cdot|x) = \mathcal{N}(x, h I_d)$. It is also known that (FP-BH) gives a time-reversal of the heat equation at the SDE level, in the sense that we can interpret $\delta_x Q_h^{\nu,h} = \mathbf{P}_{0|h}(\cdot|Z_h = x)$. Thus $\mu_0^{\leftarrow} Q_h^{\nu,h} = \int \mathbf{P}_{0|h}(\cdot|Z_h = x)\,\mu_0^{\leftarrow}(\mathrm{d}x)$, and $\nu P_h Q_t^{\nu,h} = \nu P_{h-t}$ for all $t \in [0, h]$.

The ultimate purpose of this machinery is to affirm our earlier description of the Gibbs sampling procedure as alternating forward and backward heat flows. Indeed, notice that, if $\mu_i^X$ is the law of the iterate at some iteration $i$, then $\mu_i^X P_h$ is precisely $\mu_{i+1}^Y$ under our scheme, while $(\mu_i^X P_h)Q_h^{\pi^X,h}$ is $\mu_{i+1}^X$, assuming $Q_h^{\pi^X,h}$ is well defined for non-smooth measures $\pi^X$. Thus, while Algorithm 1 is implemented via discrete steps, it can be exactly analyzed through arguments in continuous time. We shall see the benefits of this shortly.

Instead of considering the change in metrics along the evolution of $\mu P_t$ with respect to "fixed" $\nu$, it will be useful to consider the *simultaneous* evolution of $\mu P_t, \nu P_t$ (and similarly $\mu Q_t, (\nu P_h)Q_t$). This type of computation was carried out for specific metrics in earlier work [23, 27]. The following is a more generalized form of one appearing in [57, Lemma 2]. In the lemma below, we consider an arbitrary diffusion equation with corresponding Fokker-Planck equation:

$$\mathrm{d}X_t = b_t(X_t)\,\mathrm{d}t + \mathrm{d}B_t \quad \text{and} \quad \partial_t \mu_t = -\nabla \cdot (b_t \mu_t) + \frac{1}{2}\Delta \mu_t \tag{B.1}$$

where $b_t : \mathbb{R}^d \to \mathbb{R}^d$ is smooth, $X_t \in \mathbb{R}^d$, and $\mu_t = \mathsf{Law}(X_t)$ if $X_0 \sim \mu_0$.

**Lemma 10** (Decay along forward/backward heat flows). *Let $(\mu_t)_{t \geq 0}, (\nu_t)_{t \geq 0}$ denote the laws of the solutions to (B.1) starting at $\mu_0, \nu_0$ respectively. Then, for any differentiable function $g$,*

$$\partial_t g\big(D_f(\mu_t \,\|\, \nu_t)\big) = -\frac{1}{2}\,g'\big(D_f(\mu_t \,\|\, \nu_t)\big) \times \mathbb{E}_{\mu_t}\Big\langle \nabla\big(f' \circ \frac{\mu_t}{\nu_t}\big), \nabla \log \frac{\mu_t}{\nu_t}\Big\rangle.$$

*Proof.* The case where $g \neq \mathsf{id}$ is an application of the chain rule, so it suffices to take $g = \mathsf{id}$ and simply differentiate an $f$-divergence.

For brevity, we drop the variable $x$ of functions involved, and proceed as follows:

$$\partial_t D_f(\mu_t \,\|\, \nu_t) = \int \Big\{ \big(f \circ \frac{\mu_t}{\nu_t}\big)\partial_t \nu_t + \big(f' \circ \frac{\mu_t}{\nu_t}\big)\big(\frac{\mu_t}{\nu_t}\big)' \nu_t \Big\}\,\mathrm{d}x$$

$$= \int \Big\{ \partial_t \nu_t\Big(\big(f \circ \frac{\mu_t}{\nu_t}\big) - \big(f' \circ \frac{\mu_t}{\nu_t}\big)\frac{\mu_t}{\nu_t}\Big) + \big(f' \circ \frac{\mu_t}{\nu_t}\big)\partial_t \mu_t \Big\}\,\mathrm{d}x$$

$$\overset{(i)}{=} \int \Big[-\nabla \cdot (b_t \nu_t) + \frac{1}{2}\Delta \nu_t\Big]\Big(\big(f \circ \frac{\mu_t}{\nu_t}\big) - \big(f' \circ \frac{\mu_t}{\nu_t}\big)\frac{\mu_t}{\nu_t}\Big)\,\mathrm{d}x$$

$$+ \int \Big[-\nabla \cdot (b_t \mu_t) + \frac{1}{2}\Delta \mu_t\Big]\big(f' \circ \frac{\mu_t}{\nu_t}\big)\,\mathrm{d}x,$$

where in $(i)$ we substitute the F-P equation from (B.1). Integrating by parts (i.e., $\int f \operatorname{div}(\mathbf{G}) = -\int \langle \nabla f, \mathbf{G}\rangle$ for a real-valued function $f$ and vector-valued function $\mathbf{G}$), we have that

$$\int \big[-\nabla \cdot (b_t \nu_t)\big]\big(f \circ \frac{\mu_t}{\nu_t}\big)\,\mathrm{d}x = \int \Big\langle b_t \nu_t, \big(f' \circ \frac{\mu_t}{\nu_t}\big)\nabla \frac{\mu_t}{\nu_t}\Big\rangle\,\mathrm{d}x. \tag{B.2}$$

On the other hand, we have that

$$-\int \big[-\nabla \cdot (b_t \nu_t)\big]\big(f' \circ \frac{\mu_t}{\nu_t}\big)\frac{\mu_t}{\nu_t}\,\mathrm{d}x = -\int \Big\langle b_t \nu_t, \frac{\mu_t}{\nu_t}\nabla\big(f' \circ \frac{\mu_t}{\nu_t}\big) + \big(f' \circ \frac{\mu_t}{\nu_t}\big)\nabla \frac{\mu_t}{\nu_t}\Big\rangle\,\mathrm{d}x.$$

The second term cancels with the RHS of (B.2). We have a similar cancellation for the $\frac{1}{2}\Delta\nu_t$ term:

$$\int \frac{1}{2}\Delta\nu_t \left(f \circ \frac{\mu_t}{\nu_t}\right) \mathrm{d}x = -\int \frac{1}{2}\left\langle \nabla\nu_t, \left(f' \circ \frac{\mu_t}{\nu_t}\right)\nabla\frac{\mu_t}{\nu_t}\right\rangle \mathrm{d}x\,,$$

and

$$-\int \frac{1}{2}\Delta\nu_t \left(f' \circ \frac{\mu_t}{\nu_t}\right)\frac{\mu_t}{\nu_t} \mathrm{d}x = \int \frac{1}{2}\left\langle \nabla\nu_t, \frac{\mu_t}{\nu_t}\nabla\left(f' \circ \frac{\mu_t}{\nu_t}\right) + \left(f' \circ \frac{\mu_t}{\nu_t}\right)\nabla\frac{\mu_t}{\nu_t}\right\rangle \mathrm{d}x\,.$$

Combining these, we are left with

$$\int \left[-\nabla \cdot (b_t\nu_t) + \frac{1}{2}\Delta\nu_t\right]\left(\left(f \circ \frac{\mu_t}{\nu_t}\right) - \left(f' \circ \frac{\mu_t}{\nu_t}\right)\frac{\mu_t}{\nu_t}\right) \mathrm{d}x$$

$$= -\int \left\langle b_t\nu_t - \frac{1}{2}\nabla\nu_t, \nabla\left(f' \circ \frac{\mu_t}{\nu_t}\right)\frac{\mu_t}{\nu_t}\right\rangle \mathrm{d}x$$

$$= -\int \left\langle b_t\mu_t - \frac{1}{2}\mu_t\nabla\log\nu_t, \nabla\left(f' \circ \frac{\mu_t}{\nu_t}\right)\right\rangle \mathrm{d}x\,.$$

Finally, we note that

$$\int \left[-\nabla \cdot (b_t\mu_t) + \frac{1}{2}\Delta\mu_t\right]\left(f' \circ \frac{\mu_t}{\nu_t}\right) \mathrm{d}x = \int \left\langle b_t\mu_t - \frac{1}{2}\nabla\mu_t, \nabla\left(f' \circ \frac{\mu_t}{\nu_t}\right)\right\rangle \mathrm{d}x$$

$$= \int \left\langle b_t\mu_t - \frac{1}{2}\mu_t\nabla\log\mu_t, \nabla\left(f' \circ \frac{\mu_t}{\nu_t}\right)\right\rangle \mathrm{d}x\,.$$

Putting it all together, noticing that the drift terms cancel, we are left with

$$\partial_t D_f(\mu_t \,\|\, \nu_t) = -\int \frac{1}{2}\left\langle \mu_t\nabla\log\frac{\mu_t}{\nu_t}, \nabla\left(f' \circ \frac{\mu_t}{\nu_t}\right)\right\rangle \mathrm{d}x = -\frac{1}{2}\mathbb{E}_{\mu_t}\left\langle \nabla\log\frac{\mu_t}{\nu_t}, \nabla\left(f' \circ \frac{\mu_t}{\nu_t}\right)\right\rangle\,,$$

which completes the proof. $\qquad\square$

To recover the decay result for the $q$-Rényi divergence, one can substitute $g(x) = \frac{1}{q-1}\log x$ and $f(x) = x^q - 1$. For the $\chi^2$-divergence, instead substitute $g(x) = x$ and $f(x) = x^2 - 1$. From this, we can obtain a single step of decay for the Rényi and $\chi^2$-divergences under different functional inequalities.

Before proceeding, we need a standard lemma on functional inequalities under (FH).

**Lemma 11** (Functional inequalities under Gaussian convolutions, [58, Corollary 13]). *The following inequality holds for any $\pi$ with finite log-Sobolev and Poincaré constants,*

$$C_{\mathsf{PI}}(\pi P_t) \le C_{\mathsf{PI}}(\pi) + t\,, \qquad \text{and} \qquad C_{\mathsf{LSI}}(\pi P_t) \le C_{\mathsf{LSI}}(\pi) + t\,.$$

Combining the previous two lemmas, we can establish contraction between $\mu P_h Q_h$ and $\nu$ after one forward/backward iteration.

**Theorem 4** ([27, Theorem 3 and 4]). *Assume $\nu$, a measure with smooth density, satisfies* (LSI-I) *with constant $C_{\mathsf{LSI}}$. For any $q \ge 1$ and initial distribution $\mu$ with a smooth density, denoting again $Q_h := Q_h^{\nu,h}$,*

$$\mathcal{R}_q(\mu P_h Q_h \,\|\, \nu) \le \frac{\mathcal{R}_q(\mu \,\|\, \nu)}{(1 + h/C_{\mathsf{LSI}})^{2/q}}\,.$$

*If $\nu$ satisfies* (PI) *with constant $C_{\mathsf{PI}}$, then it follows that*

$$\chi^2(\mu P_h Q_h \,\|\, \nu) \le \frac{\chi^2(\mu \,\|\, \nu)}{(1 + h/C_{\mathsf{PI}})^2}\,.$$

*Moreover, for all $q \ge 2$,*

$$\mathcal{R}_q(\mu P_h Q_h \,\|\, \nu) \le \begin{cases} \mathcal{R}_q(\mu \,\|\, \nu) - \frac{2\log(1+h/C_{\mathsf{PI}})}{q} & \text{if } \mathcal{R}_q(\mu \,\|\, \nu) \ge 1\,, \\ \frac{\mathcal{R}_q(\mu\|\nu)}{(1+h/C_{\mathsf{PI}})^{2/q}} & \text{if } \mathcal{R}_q(\mu \,\|\, \nu) \le 1\,. \end{cases}$$

*Proof.* Since the SDE in (B.1) captures the forward heat flow (FH), we set $\mu_0$ and $\nu_0$ in Lemma 10 to $\mu$ and $\nu$, respectively, obtaining contraction along the forward heat flow as follows: Substituting the $q$-Rényi into Lemma 10, we have, from the definition of the Rényi divergence as $\mathcal{R}_q(\mu \parallel \nu) := \frac{1}{q-1}\log(D_f(\mu \parallel \nu) + 1)$, with $f(x) = x^q - 1$ and $g(x) = \frac{1}{q-1}\log(x+1)$,

$$\partial_t \mathcal{R}_q(\mu P_t \parallel \nu P_t) = -\frac{q}{2}\frac{\mathbb{E}_{\mu P_t}\left[\left\langle \nabla\left(\frac{\mu P_t}{\nu P_t}\right)^{q-1}, \nabla \log \frac{\mu P_t}{\nu P_t}\right\rangle\right]}{(q-1)\,\mathbb{E}_{\nu P_t}\left[\left(\frac{\mu P_t}{\nu P_t}\right)^q\right]}$$

$$\underset{(i)}{=} -\frac{q}{2}\frac{\mathbb{E}_{\mu P_t}\left[\left(\frac{\mu P_t}{\nu P_t}\right)^{q-2}\langle\nabla\frac{\mu P_t}{\nu P_t}, \nabla \log \frac{\mu P_t}{\nu P_t}\rangle\right]}{\mathbb{E}_{\nu P_t}\left[\left(\frac{\mu P_t}{\nu P_t}\right)^q\right]}$$

$$\underset{(ii)}{=} -\frac{q}{2}\frac{\mathbb{E}_{\nu P_t}\left[\left(\frac{\mu P_t}{\nu P_t}\right)^q\|\nabla \log \frac{\mu P_t}{\nu P_t}\|^2\right]}{\mathbb{E}_{\nu P_t}\left[\left(\frac{\mu P_t}{\nu P_t}\right)^q\right]} = -\frac{1}{2}\,\mathsf{RF}_q(\mu P_t \parallel \nu P_t),$$

where in $(i)$, we use again that $\nabla\left[f'\left(\frac{\mu_t}{\nu_t}\right)\frac{\mu_t}{\nu_t}\right] = \nabla\left(f' \circ \frac{\mu_t}{\nu_t}\right) \cdot \frac{\mu_t}{\nu_t} + f'\left(\frac{\mu_t}{\nu_t}\right)\nabla\frac{\mu_t}{\nu_t}$, and $(ii)$ uses that $\nabla\frac{\mu P_t}{\nu P_t} = \frac{\mu P_t}{\nu P_t}\nabla \log \frac{\mu P_t}{\nu P_t}$, and the last equality recalls the definition of the Rényi Fisher information. This yields

$$\partial_t \mathcal{R}_q(\mu P_t \parallel \nu P_t) = -\frac{1}{2}\mathsf{RF}_q(\mu P_t \parallel \nu P_t) \underset{(i)}{\leq} -\frac{1}{q}\frac{\mathcal{R}_q(\mu P_t \parallel \nu P_t)}{C_{\mathsf{LSI}}(\nu P_t)} \underset{(ii)}{\leq} -\frac{1}{q}\frac{\mathcal{R}_q(\mu P_t \parallel \nu P_t)}{C_{\mathsf{LSI}} + t},$$

where we used Lemma 8 in $(i)$ and Lemma 11 in $(ii)$. Applying Grönwall's inequality (Lemma 5),

$$\mathcal{R}_q(\mu P_h \parallel \nu P_h) \leq \exp\left(-\frac{1}{q}\int_0^h \frac{1}{C_{\mathsf{LSI}} + t}\,\mathrm{d}t\right)\mathcal{R}_q(\mu \parallel \nu) \leq \frac{\mathcal{R}_q(\mu \parallel \nu)}{(1 + h/C_{\mathsf{LSI}})^{1/q}}.$$

Since the SDE (B.1) also captures the backward equation (BH), we set $\mu_0$ and $\nu_0$ in Lemma 10 to $\mu P_h$ and $\tilde{\nu} := \nu P_h$ respectively, obtaining contraction along the backward heat flow:

$$\partial_t \mathcal{R}_q(\mu P_h Q_t \parallel \tilde{\nu}Q_t) = -\frac{1}{2}\,\mathsf{RF}_q(\mu P_h Q_t \parallel \tilde{\nu}Q_t)$$

$$\leq -\frac{1}{q}\frac{\mathcal{R}_q(\mu P_h Q_t \parallel \tilde{\nu}Q_t)}{C_{\mathsf{LSI}}(\tilde{\nu}Q_t)} \underset{(i)}{\leq} -\frac{1}{q}\frac{\mathcal{R}_q(\mu P_h Q_t \parallel \tilde{\nu}Q_t)}{C_{\mathsf{LSI}} + h - t},$$

where $(i)$ follows from that $\tilde{\nu}Q_t = \nu P_h Q_t = \nu P_{h-t}$ and $C_{\mathsf{LSI}}(\tilde{\nu}Q_t) \leq C_{\mathsf{LSI}} + h - t$ due to Lemma 11. Applying Lemma 5 again yields

$$\mathcal{R}_q(\mu P_h Q_h \parallel \nu) \leq \frac{\mathcal{R}_q(\mu P_h \parallel \tilde{\nu})}{(1 + h/C_{\mathsf{LSI}})^{1/q}}.$$

Composing these two inequalities leads to the decay rate claimed in the theorem.

The result in the $\chi^2$-divergence can be derived entirely analogously. For instance, the decay from the forward part can be shown as follows:

$$\partial_t \chi^2(\mu P_t \parallel \nu P_t) = -\frac{1}{2}\,\mathbb{E}_{\nu P_t}\left[\left\|\nabla\frac{\mu P_t}{\nu P_t}\right\|^2\right] \underset{(i)}{\leq} -\frac{\chi^2(\mu P_t \parallel \nu P_t)}{C_{\mathsf{PI}}(\nu P_t)} \leq -\frac{\chi^2(\mu P_t \parallel \nu P_t)}{C_{\mathsf{PI}} + t},$$

where $(i)$ follows from Lemma 6. Applying Grönwall's inequality then gives

$$\chi^2(\mu P_h \parallel \nu P_h) \leq \exp\left(-\int_0^h \frac{1}{C_{\mathsf{PI}} + t}\,\mathrm{d}t\right)\chi^2(\mu \parallel \nu) \leq \frac{\chi^2(\mu \parallel \nu)}{1 + h/C_{\mathsf{PI}}}.$$

The decay along the backward heat flow in $\chi^2$ is entirely analogous to the Rényi case. Then we combine two contraction results from the forward and backward flows, completing the proof.

The result in the $\mathcal{R}_q$ under (PI) can be shown in a similar manner. Only difference is that in forward and backward computations, one should use the functional inequality in Lemma 7 and the following standard inequalities:

$$1 - \exp\left(-\mathcal{R}_q(\mu \parallel \nu)\right) \geq \begin{cases} \frac{1}{2} & \text{if } \mathcal{R}_q(\mu \parallel \nu) \geq 1, \\ \frac{1}{2}\mathcal{R}_q(\mu \parallel \nu) & \text{if } \mathcal{R}_q(\mu \parallel \nu) \leq 1. \end{cases}$$

$\square$

**Part II: Extension to constrained distributions.** We now prove a technical lemma that extends the contraction results to constrained distributions. This lemma guarantees the existence of weak solutions to two stochastic processes that describe the evolution of distributions involved in Line 2 and 3 in In-and-Out, in addition to lower-semicontinuity of $f$-divergence. We shall prove it for any measure that is absolutely continuous with respect to $\pi^X$, since this imposes no additional technical hurdles.

**Lemma 12.** *Let $\nu$ be a measure, absolutely continuous with respect to the uniform measure $\pi^X$. The forward and backward heat flow equations given by*

$$\partial_t \mu_t = \frac{1}{2}\Delta\mu_t \,,$$

$$\partial_t \mu_t^\leftarrow = -\operatorname{div}\big(\mu_t^\leftarrow \nabla\log(\nu P_{h-t})\big) + \frac{1}{2}\Delta\mu_t^\leftarrow \quad \text{with } \mu_0^\leftarrow = \mu_h \,,$$

*admit solutions on $(0, h]$, and the weak limit $\lim_{t\to h}\mu_t^\leftarrow = \mu_h^\leftarrow$ exists for any initial measure $\mu_0$ with bounded support. Moreover, for any $f$-divergence with $f$ lower semi-continuous,*

$$D_f(\mu_h^\leftarrow \,\|\, \nu) \le \lim_{t\downarrow 0} D_f(\mu_{h-t}^\leftarrow \,\|\, \nu_t)\,.$$

*Proof.* The existence of weak solutions for the forward equation is well-known, since $\mu_0$ can be weakly approximated by measures with continuous density, for which the heat equation admits a unique solution for all time. In particular, the weak solution is $C^\infty$ for $t > 0$.

The reverse SDE is more subtle, since $\nabla\log\nu P_t$ will in general cease to be Lipschitz as $t \to 0$. On the other hand, for any $h > 0$, we can write explicitly

$$\mu_h(x) = \frac{1}{(2\pi h)^{d/2}}\int \exp\big(-\frac{\|x-y\|^2}{2h}\big)\,\mathrm{d}\mu_0(y)\,.$$

If one considers the system started at $\tilde\mu_0 = \mu_\epsilon = \nu P_\epsilon$ and solve the forward-backward Fokker-Planck equations on times $[0, h-\epsilon]$, then $\tilde\mu_{h-\epsilon} = \mu_h = \mu_0^\leftarrow = \tilde\mu_0^\leftarrow$ and

$$\mu_{h-\epsilon}^\leftarrow(x) = \tilde\mu_{h-\epsilon}^\leftarrow(x) = \int \frac{\exp\big(-\frac{\|x-y\|^2}{2(h-\epsilon)}\big)\nu P_\epsilon(x)}{\int \exp\big(-\frac{\|z-y\|^2}{2(h-\epsilon)}\big)\nu P_\epsilon(z)\,\mathrm{d}z}\,\mathrm{d}\mu_h(y)\,.$$

This follows from that if we consider system started at time $\epsilon > 0$, with initial distribution $\mu_\epsilon$, then we obtain the above through the Bayesian perspective on the forward and reverse heat semigroups, elaborated in Appendix E.

We now show that the following integral is indeed integrable, so $\tilde\mu_h^\leftarrow$ is well-defined:

$$\tilde\mu_h^\leftarrow(x) := \int \frac{\exp\big(-\frac{\|x-y\|^2}{2h}\big)\nu(x)}{\int \exp\big(-\frac{\|z-y\|^2}{2h}\big)\nu(z)\,\mathrm{d}z}\,\mathrm{d}\mu_h(y)\,.$$

For fixed $x$ and $\epsilon < h/2$,

$$\int \exp\big(-\frac{\|z-y\|^2}{2(h-\epsilon)}\big)\nu(z)\,\mathrm{d}z \gtrsim \exp\big(-\frac{(\|y-x_0\|+D)^2}{2(h-\epsilon)}\big)\,,$$

as the support of $\nu$ is constrained to $\mathcal{K} \subset B_D(x_0)$. Since $\mu_0$ has bounded support, $\mu_h(y) \lesssim \exp(-\frac{\|y\|^2}{a})$ for some constant $a > 0$. Thus,

$$\frac{\exp\big(-\frac{\|x-y\|^2}{2(h-\epsilon)}\big)\mu_h(y)}{\int \exp\big(-\frac{\|z-y\|^2}{2(h-\epsilon)}\big)\nu P_\epsilon(z)\,\mathrm{d}z} \lesssim \frac{\exp\big(-\frac{\|x-y\|^2}{2(h-\epsilon)}\big)\mu_h(y)}{\exp\big(-\frac{(\|y-x_0\|+D)^2}{2(h-\epsilon)}\big)}$$

$$\lesssim \exp\Big(\frac{\langle 2(x-x_0), y\rangle + 2D\|y-x_0\|}{h} - \frac{\|y\|^2}{a}\Big)\,,$$

and the last bound is integrable in $y$.

We then show the pointwise convergence of $\tilde{\mu}_{h-\epsilon}^{\leftarrow}$ to $\tilde{\mu}_h^{\leftarrow}$ as $\epsilon \to 0$. Note that $\nu P_\epsilon \to \nu$, as $\nu$ has a bounded support. Also, the denominator is independent of $\epsilon$ due to

$$\frac{1}{(2\pi(h-\epsilon))^{d/2}} \int \exp\left(-\frac{\|z-y\|^2}{2(h-\epsilon)}\right) \nu P_\epsilon(z) \, \mathrm{d}z = \mathcal{N}(0, (h-\epsilon)I) * \nu P_\epsilon = \nu * \mathcal{N}(0, hI) \, .$$

Hence, for $\epsilon \leq d^{-1}$,

$$\frac{\exp\left(-\frac{\|x-y\|^2}{2(h-\epsilon)}\right)}{\int \exp\left(-\frac{\|z-y\|^2}{2(h-\epsilon)}\right) \nu P_\epsilon(z) \, \mathrm{d}z} \leq \left(\frac{h}{h-\epsilon}\right)^{d/2} \frac{\exp\left(-\frac{\|x-y\|^2}{2h}\right)}{\int \exp\left(-\frac{\|z-y\|^2}{2h}\right) \nu(z) \, \mathrm{d}z}$$

$$\lesssim \frac{\exp\left(-\frac{\|x-y\|^2}{2h}\right)}{\int \exp\left(-\frac{\|z-y\|^2}{2h}\right) \nu(z) \, \mathrm{d}z} \, .$$

As shown above, the last bound is integrable with respect to $\mu_h$, so the dominated convergence theorem implies

$$\lim_{\epsilon \to 0} \int \frac{\exp\left(-\frac{\|x-y\|^2}{2(h-\epsilon)}\right)}{\int \exp\left(-\frac{\|z-y\|^2}{2(h-\epsilon)}\right) \nu P_\epsilon(z) \, \mathrm{d}z} \, \mathrm{d}\mu_h(y) = \int \frac{\exp\left(-\frac{\|x-y\|^2}{2h}\right)}{\int \exp\left(-\frac{\|z-y\|^2}{2h}\right) \nu(z) \, \mathrm{d}z} \, \mathrm{d}\mu_h(y) \, ,$$

Thus, the pointwise convergence follows. Note that if we take $\nu(x) = \pi^X(x) = \frac{\mathbb{1}_{\mathcal{K}}(x)}{\mathrm{vol}(\mathcal{K})}$, then $\tilde{\mu}_h^{\leftarrow}$ is the distribution of the backwards step of our algorithm. In particular, this corresponds to first sampling $x \sim \mu_h$, then $y \sim Q_h^{\nu,h}(\cdot|x)$, which is precisely the law of $\mu_h^{\leftarrow}$ given by (FP-BH).

As for the second statement, it follows from Scheffé's lemma [59, Theorem 16.12] that the pointwise convergence of $\mu_{h-\varepsilon}^{\leftarrow} \to \mu_h^{\leftarrow}$ leads to its TV-convergence, which in turn implies the weak convergence. It follows from lower semicontinuity of $D_f$ [60, Theorem 2.34] that the weak convergence ensures $D_f(\mu_h^{\leftarrow} \| \nu) \leq \lim_{t \downarrow 0} D_f(\mu_{h-t}^{\leftarrow} \| \nu_{h-t}^{\leftarrow})$. $\qquad \square$

In the sequel, we will only consider $\nu = \pi^X$. Since the Rényi divergence is a continuous function of the $\chi^q$ divergence (see Definition 1), which itself is an $f$-divergence, it enjoys the same lower-semicontinuity properties. Using this lower-semicontinuity together with the decay results in Theorem 4, we can easily derive the contraction results of In-and-Out in $\mathcal{R}_q$ and $\chi^q$ for any $q \geq 1$. We remark that this result does not require convexity of $\mathcal{K}$.

*Proof of Theorem 3.* Let us set $\mu_0 = \mu_0^X$ and $\pi_0 = \pi^X$. Then, $\mu_h = \mu_0^{\leftarrow} = \mu_1^Y$, $\pi_h = \pi_0^{\leftarrow} = \pi^Y$, and $\mu_h^{\leftarrow} = \mu_1^X$, $\pi_h^{\leftarrow} = \pi^X$. For small $\epsilon > 0$, as $\mu_\epsilon = (\mu_0^X)_\epsilon = \mu_0^X * \mathcal{N}(0, \epsilon I_d)$ is $C^\infty$-smooth, we can now invoke the decay results with step size $h - \epsilon$ in Theorem 4. Thus, for contraction constants $C_\epsilon = (1 + \frac{h-\epsilon}{C_{\mathsf{LSI}}+\epsilon})^{-2/q}$ and $C_\epsilon = (1 + \frac{h-\epsilon}{C_{\mathsf{PI}}+\epsilon})^{-2}$ respectively when $\Phi = \mathcal{R}_q$ and $\Phi = \chi^2$,

$$\Phi(\mu_{h-\epsilon}^{\leftarrow} \| \pi_\epsilon) \leq C_\epsilon \cdot \Phi(\mu_\epsilon \| \pi_\epsilon) \leq C_\epsilon \cdot \Phi(\mu_0 \| \pi_0) \, ,$$

where we used the data-processing inequality for the last inequality. By the second result of Lemma 12, sending $\epsilon \to 0$ leads to

$$\Phi(\mu_1^X \| \pi^X) = \Phi(\mu_h^{\leftarrow} \| \pi_0) \leq C \cdot \Phi(\mu_0 \| \pi_0) = C \cdot \Phi(\mu_0^X \| \pi^X) \, .$$

Repeating this argument $k$ times completes the proof. $\qquad \square$

### B.3  Failure probability and wasted steps

We begin by defining a suitable version of *local conductance* [4].

**Definition 6** (Local conductance). The local conductance $\ell$ on $\mathbb{R}^d$ is defined by

$$\ell(x) \stackrel{\text{def}}{=} \frac{\int_{\mathcal{K}} \exp(-\frac{1}{2h}\|x-y\|^2) \, \mathrm{d}y}{\int_{\mathbb{R}^d} \exp(-\frac{1}{2h}\|x-y\|^2) \, \mathrm{d}y} = \frac{\int_{\mathcal{K}} \exp(-\frac{1}{2h}\|x-y\|^2) \, \mathrm{d}y}{(2\pi h)^{d/2}} \, .$$

The local conductance at $y$ quantifies the success probability of the proposal at $y$ in Line 3. Then the expected number of trials until the first success of Line 3 is $1/\ell(y)$. Revisiting (3.1), we can notice $\pi^Y(y) = \ell(y)/\mathrm{vol}(\mathcal{K})$.

**Naïve analysis for expected number of trials.** Starting from $\pi^X$, when we just naïvely sample from $\pi^{Y|X}(\cdot|x)$ for all $x$ without imposing any *failure* condition, the expected number of trials for one iteration is that for the probability density $p_x$ of $\mathcal{N}(x, hI_d)$,

$$\int_{\mathcal{K}} \int_{\mathbb{R}^d} \frac{1}{\ell(y)} \, p_x(\mathrm{d}y) \pi^X(\mathrm{d}x) = \int_{\mathbb{R}^d} \frac{1}{\ell(y)} \, \pi^Y(\mathrm{d}y) = \int_{\mathbb{R}^d} \frac{1}{\ell(y)} \frac{\ell(y)}{\mathrm{vol}(\mathcal{K})} \, \mathrm{d}y = \infty \,.$$

This suggests that one should consider the algorithm as having "failed" if the number of trials exceeds some threshold.

**Refined analysis under a failure condition.** Going forward, we assume an $M$-warm start as in previous work for uniform sampling algorithms. By induction we have $\frac{\mathrm{d}\mu_i^X}{\mathrm{d}\pi^X} \leq M$ for all $i$.

**Lemma 13** (Propagation of warm-start). *From an $M$-warm start, we have $\mathrm{d}\mu_i^X/\mathrm{d}\pi^X \leq M$ for all $i$.*

*Proof.* Assume that $\mu_i^X$ satisfies the $M$-warm start. Then, for any measurable $S$ and the transition kernel $T_x$ of Algorithm 1 at $x$,

$$\mu_{i+1}^X(S) = \int_{\mathcal{K}} T_x(S) \, \mathrm{d}\mu_i^X(x) \leq M \int_{\mathcal{K}} T_x(S) \, \mathrm{d}\pi^X(x) = M\pi^X(S) \,,$$

where the last equality follows from the stationarity of $\pi$. Hence, $\mathrm{d}\mu_{i+1}^X/\mathrm{d}\pi^X \leq M$. $\qquad\square$

We now establish a lemma that comes in handy when analyzing the failure probability of the algorithm. In essence, this lemma bounds the probability that taking a Gaussian step from $\pi^X$ in Line 2 gets $\delta$-distance away from $\mathcal{K}$. Let us denote the $\delta$-blowup of $\mathcal{K}$ by $\mathcal{K}_\delta := \{x \in \mathbb{R}^d : d(x, \mathcal{K}) \leq \delta\}$.

**Lemma 14.** *For a convex body $\mathcal{K} \subset \mathbb{R}^d$ containing a unit ball $B_1(0)$,*

$$\pi^Y(\mathcal{K}_\delta^c) \leq \exp\left(-\frac{\delta^2}{2h} + \delta d\right) \,.$$

*Proof.* For $y \in \partial\mathcal{K}_\delta$, we can take the supporting half-space $H(y)$ at $\mathrm{proj}_{\mathcal{K}}(y)$ containing $\mathcal{K}$, due to convexity of $\mathcal{K}$. Then,

$$\begin{aligned}
\pi^Y(\mathcal{K}_\delta^c) &= \frac{1}{\mathrm{vol}(\mathcal{K})} \int_{\mathcal{K}_\delta^c} \int_{\mathcal{K}} \frac{\exp(-\frac{1}{2h}\|y-x\|^2)}{(2\pi h)^{d/2}} \, \mathrm{d}x \, \mathrm{d}y \\
&\leq \frac{1}{\mathrm{vol}(\mathcal{K})} \int_{\mathcal{K}_\delta^c} \int_{H(y)} \frac{\exp(-\frac{1}{2h}\|y-x\|^2)}{(2\pi h)^{d/2}} \, \mathrm{d}x \, \mathrm{d}y \\
&= \frac{1}{\mathrm{vol}(\mathcal{K})} \int_{\mathcal{K}_\delta^c} \int_{d(y,\mathcal{K})}^{\infty} \frac{\exp(-\frac{z^2}{2h})}{\sqrt{2\pi h}} \, \mathrm{d}z \, \mathrm{d}y \,.
\end{aligned} \tag{B.3}$$

Let us denote the tail probability of the 1-dimensional Gaussian with variance $h$ by

$$\mathsf{T}(s) := \mathbb{P}_{\mathcal{N}(0,h)}(Z \geq s) = 1 - \Phi(h^{-1/2}s) \,,$$

where $\Phi$ is the CDF of the standard Gaussian. By the co-area formula and integration by parts,

$$\begin{aligned}
\int_{\mathcal{K}_\delta^c} \int_{d(y,\mathcal{K})}^{\infty} \frac{\exp(-\frac{1}{2h}z^2)}{\sqrt{2\pi h}} \, \mathrm{d}z\mathrm{d}y &= \int_{\delta}^{\infty} \mathsf{T}(s) \, \mathrm{vol}(\partial\mathcal{K}_s) \, \mathrm{d}s \\
&= \underbrace{\left[\mathsf{T}(s) \int_0^s \mathrm{vol}(\partial\mathcal{K}_z) \, \mathrm{d}z\right]_\delta^{\infty}}_{=:F} + \int_{\delta}^{\infty} \frac{1}{\sqrt{2\pi h}} \exp\left(-\frac{s^2}{2h}\right) \int_0^s \mathrm{vol}(\partial\mathcal{K}_z) \, \mathrm{d}z \, \mathrm{d}s \,.
\end{aligned} \tag{B.4}$$

Recall that $\mathsf{T}(s) \leq \frac{1}{2}\exp(-\frac{1}{2}(h^{-1/2}s)^2)$ for $h^{-1/2}s \geq 0$ due to a standard tail bound on a Gaussian distribution. This tail bound, combined with

$$\int_0^s \mathrm{vol}(\partial\mathcal{K}_z) \, \mathrm{d}z = \mathrm{vol}(\mathcal{K}_s) - \mathrm{vol}(\mathcal{K}) \leq \mathrm{vol}\big((1+s)\,\mathcal{K}\big) - \mathrm{vol}(\mathcal{K}) = \big((1+s)^d - 1\big) \mathrm{vol}(\mathcal{K}) \,,$$

ensures that F vanishes at $s = \infty$. Hence, bounding the first term in (B.4) by 0 results in

$$\int_{\mathcal{K}_\delta^c} \int_{d(y,\mathcal{K})}^\infty \frac{\exp(-\frac{1}{2h}z^2)}{\sqrt{2\pi h}} \, \mathrm{d}z \, \mathrm{d}y \leq \frac{1}{\sqrt{2\pi h}} \int_\delta^\infty \exp\left(-\frac{s^2}{2h}\right) \underbrace{\left((1+s)^d - 1\right)}_{\leq \exp(sd)} \mathrm{vol}(\mathcal{K}) \, \mathrm{d}s$$

$$\leq \frac{\mathrm{vol}(\mathcal{K})}{\sqrt{2\pi h}} \exp(hd^2/2) \int_\delta^\infty \exp\left(-\frac{1}{2h}(s - hd)^2\right) \mathrm{d}s$$

$$\underset{(i)}{\leq} \mathrm{vol}(\mathcal{K}) \exp(hd^2/2) \exp\left(-\frac{(\delta - hd)^2}{2h}\right)$$

$$= \mathrm{vol}(\mathcal{K}) \exp\left(-\frac{\delta^2}{2h} + \delta d\right),$$

where in $(i)$ we used the tail bound for a Gaussian. $\qquad\square$

This core lemma suggests taking $\delta = t/d$ and $h = c/d^2$ for some $t, c > 0$, under which we have

$$\pi^Y(\mathcal{K}_\delta^c) \leq \exp\left(-\frac{t^2}{2c} + t\right).$$

Now we choose a suitable threshold $N$ for bounding the failure probability. Following (B.3) in the proof, one can notice that for $y \in \mathcal{K}_\delta^c$, $\delta = \Omega(1/d)$, and $h = \Theta(d^{-2})$,

$$\ell(y) \leq \int_{d(y,\mathcal{K})}^\infty \frac{\exp(-\frac{1}{2h}z^2)}{\sqrt{2\pi h}} \, \mathrm{d}z = \mathbb{P}_{Z \sim \mathcal{N}(0,h)}(Z \geq \delta) \leq \exp(-\Omega(t^2)).$$

Thus, the expected number of trials from $\mathcal{K}_\delta^c$ for the rejection sampling in Line 3 is $\ell(y)^{-1} \geq \exp(\Omega(t^2))$. Intuitively, one can ignore whatever happens in $\mathcal{K}_\delta^c$, since $\mathcal{K}_\delta$ takes up most of measure of $\pi^Y$. As the number of trials from $\mathcal{K}_\delta^c$ is at least $\exp(\Omega(t^2))$ in expectation, the most straightforward way to ignore algorithmic behaviors from $\mathcal{K}_\delta^c$ is simply to set the threshold to $N = \widetilde{\mathcal{O}}(\exp(t^2))$. Even though the threshold is $N$, the expected number of trials is much lower.

Lemma 3 bounds the failure probability and expected number of trials per iteration.

*Proof of Lemma 3.* For $\mu_h := \mu * \mathcal{N}(0, hI_d)$, the failure probability is $\mathbb{E}_{\mu_h}[(1 - \ell)^N]$. Since $\mathrm{d}\mu/\mathrm{d}\pi^X \leq M$ implies $\mathrm{d}\mu_h/\mathrm{d}(\pi^X)_h = \mathrm{d}\mu_h/\mathrm{d}\pi^Y \leq M$, it follows that

$$\mathbb{E}_{\mu_h}[(1 - \ell)^N] \leq M \, \mathbb{E}_{\pi^Y}[(1 - \ell)^N].$$

Then,

$$\int_{\mathbb{R}^d} \underbrace{(1 - \ell)^N \, \mathrm{d}\pi^Y}_{=:A} = \int_{\mathcal{K}_\delta^c} A + \int_{\mathcal{K}_\delta \cap [\ell \geq N^{-1}\log(3mM/\eta)]} A + \int_{\mathcal{K}_\delta \cap [\ell < N^{-1}\log(3mM/\eta)]} A$$

$$\leq \pi^Y(\mathcal{K}_\delta^c) + \int_{[\ell \geq N^{-1}\log(3mM/\eta)]} \exp(-\ell N) \, \mathrm{d}\pi^Y$$

$$+ \int_{\mathcal{K}_\delta \cap [\ell < N^{-1}\log(3mM/\eta)]} \frac{\ell(y)}{\mathrm{vol}(\mathcal{K})} \, \mathrm{d}y$$

$$\leq \exp\left(-\frac{t^2}{2c} + t\right) + \frac{\eta}{3mM} + \frac{\log(3mM/\eta)}{N} \frac{\mathrm{vol}(\mathcal{K}_\delta)}{\mathrm{vol}(\mathcal{K})}$$

$$\leq \exp\left(-\frac{t^2}{2c} + t\right) + \frac{\eta}{3mM} + \frac{e^t}{N} \log \frac{3mM}{\eta},$$

where we used $\mathrm{vol}(\mathcal{K}_\delta) \subset \mathrm{vol}((1 + \delta)\mathcal{K}) = (1 + \delta)^d \mathrm{vol}(\mathcal{K}) \leq e^t \mathrm{vol}(\mathcal{K})$. Taking $c = \frac{\log\log Z}{2\log Z}$, $t = \sqrt{8}\log\log Z$, and $N = Z(\log Z)^4$, we can bound the last line by $\frac{\eta}{mM}$. Therefore,

$$\mathbb{E}_{\mu_h}[(1 - \ell(\cdot))^N] \leq M \, \mathbb{E}_{\pi^Y}[(1 - \ell(\cdot))^N] \leq \frac{\eta}{m}.$$

We now bound the expected number of trials per iteration. Let $X$ be the minimum of the threshold $N$ and the number of trials until the first success. Then the expected number of trials per step is bounded by $M \mathbb{E}_{\pi^Y} X$ since $\mathrm{d}\mu_h/\mathrm{d}\pi^Y \leq M$. Thus,

$$\int_{\mathbb{R}^d} \big(\frac{1}{\ell} \wedge N\big) \, \mathrm{d}\pi^Y \leq \int_{\mathcal{K}_\delta} \frac{1}{\ell} \, \mathrm{d}\pi^Y + N\pi^Y(\mathcal{K}_\delta^c) = \frac{\mathrm{vol}(\mathcal{K}_\delta)}{\mathrm{vol}(\mathcal{K})} + N\pi^Y(\mathcal{K}_\delta^c)$$

$$\leq e^t + N \exp\big(-\frac{t^2}{2c} + t\big) \leq (\log Z)^3 + 3(\log Z)^4 = \mathcal{O}\big(\log^4 \frac{mM}{\eta}\big).$$

Therefore, the expected number of trials per step is $\mathcal{O}(M \log^4 \frac{mM}{\eta})$, and the claim follows since each trial uses one query to the membership oracle of $\mathcal{K}$. $\qquad\square$

## B.4 Putting it together

We can now show that In-and-Out subsumes previous results on uniform sampling from convex bodies (such as Ball walk and Speedy walk), providing detailed versions of the main results in §4.

We first establish that the query complexity of In-and-Out matches that of the Ball walk under stronger divergences. Recall that $2\|\cdot\|_{\mathsf{TV}}^2 \leq \mathsf{KL} \leq \log(1 + \chi^2) \leq \chi^2$.

**Theorem 5.** *For any given $\eta, \varepsilon \in (0,1)$, $q \geq 1$, $m \in \mathbb{N}$ defined below and any convex body $\mathcal{K}$ given by a well-defined membership oracle, consider In-and-Out (Algorithm 1) with an $M$-warm initial distribution $\mu_0^X$, $h = (2d^2 \log \frac{9mM}{\eta})^{-1}$, and $N = \widetilde{\mathcal{O}}(\frac{mM}{\eta})$. For $\pi^X$ the uniform distribution over $\mathcal{K}$,*

- *It achieves $\mathcal{R}_q(\mu_m^X \| \pi^X) \leq \varepsilon$ after $m = \widetilde{\mathcal{O}}(qd^2 \|\mathrm{Cov}(\pi^X)\|_{\mathsf{op}} \log^2 \frac{M}{\eta\varepsilon})$ iterations. With probability $1 - \eta$, the algorithm iterates this many times without failure, using $\widetilde{\mathcal{O}}(qMd^2 \|\mathrm{Cov}(\pi^X)\|_{\mathsf{op}} \log^6 \frac{1}{\eta\varepsilon})$ expected number of membership queries in total.*

- *For isotropic $\pi^X$, with probability $1 - \eta$, the algorithm achieves $\mathcal{R}_q(\mu_m^X \| \pi^X) \leq \varepsilon$ with $m = \widetilde{\mathcal{O}}(qd^2 \log^2 \frac{M}{\eta\varepsilon})$ iterations, using $\widetilde{\mathcal{O}}(qMd^2 \log^6 \frac{1}{\eta\varepsilon})$ membership queries in expectation.*

*Proof.* We just put together Lemma 3 and Theorem 3. For target accuracy $\varepsilon > 0$, we use the $\mathcal{R}_q$-decay under (PI) for $q \geq 2$ in Theorem 3. The $M$-warm start assumption guarantees $\mathcal{R}_q(\mu_0^X \| \pi^X) \lesssim \log M$. Due to $C_{\mathsf{PI}}(\pi^X) = \mathcal{O}(\|\mathrm{Cov}(\pi^X)\|_{\mathsf{op}} \log d)$ (Lemma 9), In-and-Out can achieve $\mathcal{R}_q(\mu_m^X \| \pi^X) \leq \varepsilon$ after $m = \widetilde{\mathcal{O}}(qd^2 \|\mathrm{Cov}(\pi^X)\|_{\mathsf{op}} \log^2 \frac{M}{\eta\varepsilon})$ iterations. Since each iteration has $\eta/m$-failure probability by Lemma 3, the union bound ensures that the total failure probability is at most $\eta$ throughout $m$ iterations. Lastly, each iteration requires $\widetilde{\mathcal{O}}(M \log^4 \frac{1}{\eta\varepsilon})$ membership queries in expectation by Lemma 3. Therefore, In-and-Out uses $\widetilde{\mathcal{O}}(qMd^2 \min(D^2, \|\mathrm{Cov}(\pi^X)\|_{\mathsf{op}}) \log^6 \frac{1}{\eta\varepsilon})$ expected number of membership queries over $m$ iterations. Since $\mathcal{R}_q$ is non-decreasing in $q$, we can obtain the desired bound on $\mathcal{R}_q$ for $q \in [1, 2)$.

For isotropic $\pi^X$, we have $\mathrm{Cov}(\pi^X) = I_d$, so the claim immediately follows from $C_{\mathsf{PI}}(\pi^X) = \mathcal{O}(\log d)$ (see Lemma 9). $\qquad\square$

We now show that the number of proper steps is bounded as claimed for general *non-convex bodies* and *any feasible start* in $\mathcal{K}$. We first establish this result under an $M$-warm start (Theorem 2).

*Proof of Theorem 2.* By the Rényi-decay under (LSI-I) in Theorem 3, In-and-Out can achieve $\varepsilon$-distance to $\pi^X$ after $\mathcal{O}\big(qh^{-1}C_{\mathsf{LSI}}(\pi^X) \log \frac{\mathcal{R}_q(\mu_1^X \| \pi^X)}{\varepsilon}\big)$ iterations for $q \geq 1$.

For $q \geq 2$, we use the decay result under (PI). In this case, In-and-Out decays under two different rates depending on the value of $\mathcal{R}_q(\cdot \| \pi^X)$. It first needs $\mathcal{O}(qh^{-1}C_{\mathsf{PI}}(\pi^X)\mathcal{R}_q(\mu_0^X \| \pi^X))$ iterations until $\mathcal{R}_q(\cdot \| \pi^X)$ reaches 1. Then, In-and-Out additionally needs $\mathcal{O}(qh^{-1}C_{\mathsf{PI}}(\pi^X) \log \frac{1}{\varepsilon})$ iterations, and thus it needs $\mathcal{O}(qh^{-1}C_{\mathsf{PI}}(\pi^X)\big(\mathcal{R}_q(\mu_0^X \| \pi^X) + \log \frac{1}{\varepsilon}\big))$ iterations in total. By substituting $\mathcal{R}_q(\mu_0^X \| \pi^X) \lesssim \log M$, we complete the proof. $\qquad\square$

Next, we show that In-and-Out mixes from any start.

**Corollary 2.** *For any given $\varepsilon \in (0,1)$ and set $\mathcal{K} \subset B_D(0)$,* In-and-Out *with variance $h$ and any feasible start $x_0 \in \mathcal{K}$ achieves $\mathcal{R}_q(\mu_m^X \| \pi^X) \leq \varepsilon$ after $m = \widetilde{\mathcal{O}}(qh^{-1}C_{\mathsf{LSI}}(\pi^X) \log \frac{d+D^2/h}{\varepsilon})$ iterations.*

*Proof.* We first bound the warmness of $\mu_1^X$ w.r.t. $\pi^X$ when $\mu_0^X = \delta_{x_0}$. One can readily check that

$$\mu_1^X(x) = \mathbb{1}_{\mathcal{K}}(x) \cdot \int \frac{\exp\left(-\frac{1}{2h}\|y-x\|^2\right)\exp\left(-\frac{1}{2h}\|y-x_0\|^2\right)}{(2\pi h)^{d/2}\int_{\mathcal{K}}\exp\left(-\frac{1}{2h}\|y-x\|^2\right)\mathrm{d}x}\,\mathrm{d}y\,.$$

By Young's inequality, $\|y-x\|^2 \leq (\|y\|+D)^2 \leq \frac{3}{2}\|y\|^2 + 3D^2$ for $x \in \mathcal{K}$. Hence,

$$\int \frac{\exp\left(-\frac{1}{2h}\|y-x\|^2\right)\exp\left(-\frac{1}{2h}\|y-x_0\|^2\right)}{\int_{\mathcal{K}}\exp\left(-\frac{1}{2h}\|y-x\|^2\right)\mathrm{d}x}\,\mathrm{d}y$$

$$\leq \frac{\exp(2h^{-1}D^2)}{\mathrm{vol}(\mathcal{K})}\int \exp\left(-\frac{1}{2h}\left(\|y-x\|^2 + \|y-x_0\|^2 - \frac{3}{2}\|y\|^2\right)\right)\mathrm{d}y$$

$$= \frac{\exp(2h^{-1}D^2)}{\mathrm{vol}(\mathcal{K})}\int \exp\left(-\frac{1}{2h}\left(\frac{1}{2}\|y-2(x+x_0)\|^2 + (\|x\|^2 + \|x_0\|^2 - 2\|x+x_0\|^2)\right)\right)\mathrm{d}y$$

$$\leq \frac{\exp(5h^{-1}D^2)}{\mathrm{vol}(\mathcal{K})}\int \exp\left(-\frac{1}{4h}\|y-2(x+x_0)\|^2\right)\mathrm{d}y$$

$$= \frac{\exp(5h^{-1}D^2)}{\mathrm{vol}(\mathcal{K})}(4\pi h)^{d/2}\,.$$

Therefore, $M = \mathrm{ess\,sup}\,\frac{\mu_1^X}{\pi^X} \leq 2^{d/2}\exp(5h^{-1}D^2)$. By Theorem 2 under (LSI-I), In-and-Out needs $\widetilde{\mathcal{O}}(qh^{-1}C_{\mathsf{LSI}}(\pi^X) \log \frac{d+D^2/h}{\varepsilon})$ iterations. $\qquad\square$

We then obtain the following corollary for a convex body $\mathcal{K}$.

**Corollary 3.** *For any given $\varepsilon \in (0,1)$ and convex body $\mathcal{K} \subset B_D(0)$,* In-and-Out *with variance $h$ and an $M$-warm initial distribution achieves $\mathcal{R}_q(\mu_m^X \| \pi^X) \leq \varepsilon$ after $m = \widetilde{\mathcal{O}}(qh^{-1}D^2 \log \frac{1}{\varepsilon})$ iterations. If $\pi^X$ is isotropic, then* In-and-Out *only needs $\widetilde{\mathcal{O}}(qh^{-1}D \log \frac{d+d^2/h}{\varepsilon})$ iterations.*

*Proof.* For convex $\mathcal{K}$, it follows from Lemma 9 that $C_{\mathsf{LSI}}(\pi^X) = \mathcal{O}(D^2)$ and $C_{\mathsf{LSI}}(\pi^X) = \mathcal{O}(D)$ for isotropic $\mathcal{K}$. The rest of the proof can be completed in a similar way. $\qquad\square$

For $h = \widetilde{\Theta}(d^{-2})$, In-and-Out requires $\widetilde{\mathcal{O}}(qd^2D^2)$ iterations and in particular $\widetilde{\mathcal{O}}(qd^2D)$ iteration for isotropic uniform distributions. These results match those of Speedy walk [61, 37] (see Theorem 7).

## C Ball walk **and** Speedy walk

We restate the previously known guarantees for uniform sampling by Ball walk and Speedy walk. Below, let $B_r(x)$ denote the $d$-dimensional ball of radius $r$ centered at $x$.

---

**Algorithm 2** Ball walk

---

    **Input:** initial distribution $\pi_0$, convex body $\mathcal{K} \subset \mathbb{R}^d$, iterations $T$, step size $\delta > 0$.
1: Sample $x_0 \sim \pi_0$.
2: **for** $i = 1, \ldots, T$ **do**
3:     Sample $y \sim \mathrm{Unif}(B_\delta(x_{i-1}))$.
4:     If $y \in \mathcal{K}$, then $x_i \leftarrow y$. Else, $x_i \leftarrow x_{i-1}$.
5: **end for**

---

Ball walk is particularly simple; draw a uniform random point from $B_\delta$ around the current point, and go there if the drawn point is inside of $\mathcal{K}$ and stay at the current point otherwise. Its stationary distribution can be easily seen to be $\pi \propto \mathbb{1}_{\mathcal{K}}$, the uniform distribution over $\mathcal{K}$.

In the literature, there are two approaches to analyzing the convergence rate of this sampler: (i) a direct analysis via the $s$-conductance of Ball walk and (ii) an indirect approach which first passes through Speedy walk.

**Direct analysis.** The following TV-guarantee is obtained by lower bounding the $s$-conductance of Ball walk, which requires a one-step coupling argument and the Cheeger inequality for $\pi$. We refer interested readers to [16, §5].

**Theorem 6** (Convergence of Ball walk). *For any $\varepsilon \in (0, 1)$ and convex body $\mathcal{K} \subset \mathbb{R}^d$ presented by a well-defined membership oracle, let $\pi_t$ be the distribution after $t$ steps of* Ball walk *with an $M$-warm initial distribution $\pi_0$. Then,* Ball walk *with step size $\delta = \Theta(\frac{\varepsilon}{M\sqrt{d}})$ achieves $\|\pi_t - \pi\|_{\mathsf{TV}} \le \varepsilon$ for $t \gtrsim d^2 D^2 \frac{M^2}{\varepsilon^2} \log \frac{M}{\varepsilon}$. If $\pi$ is isotropic, then* Ball walk *needs $\mathcal{O}(d^2 \frac{M^2}{\varepsilon^2} \log d \log \frac{M}{\varepsilon})$ iterations.*

The mixing time of Ball walk under this approach has a polynomial dependence on $1/\varepsilon$, rather than a polylogarithmic dependence.

**Indirect analysis through** Speedy walk**.** [4] introduced Speedy walk, which could be viewed as a version of Ball walk and converges to a *speedy distribution* (see Proposition 1), which is slightly biased from $\pi$. Then, Speedy walk is used together with another algorithmic component (rejection sampling) [4, Algorithm 4.15] that converts the speedy distribution to the uniform distribution. In the literature, Ball walk often refers to 'Speedy walk combined with the conversion step', rather than a direct implementation of Algorithm 2. Strictly speaking, a mixing guarantee of this combined algorithm should not be referred to as a provable guarantee of Ball walk.

---

**Algorithm 3** Speedy walk

**Input:** initial distribution $\pi_0$, convex body $\mathcal{K} \subset \mathbb{R}^d$, iterations $T$, step size $\delta > 0$.
1: Sample $x_0 \sim \pi_0$.
2: **for** $i = 1, \dots, T$ **do**
3:   Sample $x_i \sim \text{Unif}(\mathcal{K} \cap B_\delta(x_{i-1}))$.
4: **end for**

---

As opposed to Ball walk, Speedy walk *always* takes some step at each iteration. However, the problem of sampling from $x_i \sim \text{Unif}(\mathcal{K} \cap B_\delta(x_{i-1}))$ in Line 3 is not straightforward. This step admits the following implementation based on rejection sampling, via a procedure denoted by $(*)$:

- Propose $y \sim \text{Unif}(B_\delta(x_{i-1}))$.

- Set $x_{i+1} \leftarrow y$ if $y \in \mathcal{K}$. Otherwise, repeat the proposal.

Each actual step (indexed by $i$) in Speedy walk is called a *proper step*, and rejected steps during $(*)$ are called *improper steps*. For example, if $x_1, x_1, x_2, x_3, x_3, x_3, x_4, \dots$ are the positions produced by Ball walk, then only proper steps $x_1, x_2, x_3, x_4, \dots$ are recorded by Speedy walk.

To describe the theoretical guarantees of Speedy walk, we define the *local conductance* $\ell(x)$ at $x \in \mathcal{K}$, which measures the success probability of the rejection sampling scheme in $(*)$:

$$\ell(x) := \frac{\text{vol}(\mathcal{K} \cap B_\delta(x))}{\text{vol}(B_\delta(x))},$$

and define the *average conductance*:

$$\lambda := \mathbb{E}_\pi \ell = \frac{1}{\text{vol}(\mathcal{K})} \int_\mathcal{K} \ell(x)\, \mathrm{d}x.$$

**Proposition 1** ([4]). *The stationary distribution $\nu$ of* Speedy walk *has density*

$$\nu(x) = \frac{\ell(x)\, \mathbb{1}_\mathcal{K}(x)}{\int_\mathcal{K} \ell(x)\, \mathrm{d}x}.$$

The speedy distribution $\nu$ is indeed different from the uniform distribution $\pi$, and this discrepancy is quantified in terms of the average conductance.

**Proposition 2** ([4, Page 22]). $\|\nu - \pi\|_{\mathsf{TV}} \leq \frac{1-\lambda}{\lambda}$.

One can relate the step size $\delta$ to the average conductance.

**Proposition 3** (Bound on average conductance, [4, Corollary 4.5]). $\lambda \geq 1 - \frac{\delta\sqrt{d}}{2}$.

The best known result for Speedy walk's mixing is due to [61] devising the *blocking conductance* and using the *log-Cheeger* inequality. When $\nu$ is isotropic (i.e., it has covariance proportional to the identity matrix), [37] improves the mixing bound via the *log-Cheeger* constant.

**Theorem 7** (Mixing of Speedy walk). *For any $\varepsilon \in (0,1)$ and convex body $\mathcal{K} \subset \mathbb{R}^d$ presented by a well-defined membership oracle, let $\nu_t$ be the distribution after $t$ proper steps of Speedy walk started at any feasible point $x_0 \in \mathcal{K}$. Then, Speedy walk with step size $\delta = \Theta(d^{-1/2})$ achieves $\|\nu_t - \nu\|_{\mathsf{TV}} \leq \varepsilon$ for $t \gtrsim (D^2 + \log(D\sqrt{d}))\, d^2 \log\frac{1}{\varepsilon}$. From an $M$-warm start, the expected number of improper steps during $t$ iterations is $\widetilde{\mathcal{O}}(tM)$. When $\nu$ is isotropic, Speedy walk needs $\mathcal{O}(d^2 D \log\frac{1}{\varepsilon} \log\log D)$ proper steps to achieve $\varepsilon$-TV distance to $\nu$.*

Then, [4] uses the following post-processing step to obtain an approximately uniform distribution on $\mathcal{K}$, with a provable guarantee.

$\quad$ $\mathcal{A}$: Call Speedy walk to obtain a sample $X \sim \nu_t$ until $\frac{2d}{2d-1}\, X \in \mathcal{K}$. If so, return $\bar{X} = \frac{2d}{2d-1}\, X$.

**Proposition 4** ([4, Theorem 4.16]). *Under the same setting above, assume $\|\nu_t - \nu\|_{\mathsf{TV}} \leq \varepsilon$ for step size $\delta \leq (8d\log\frac{d}{\varepsilon})^{-1/2}$ and fixed $t \in \mathbb{N}$. For $\bar{\nu} = \mathrm{law}(\bar{X})$ given by $\mathcal{A}$, it holds that $\|\bar{\nu} - \pi\|_{\mathsf{TV}} \leq \varepsilon$, and the expected number of calls on the conversion algorithm is at most 2.*

Combining the previous two results, we conclude that the total expected number of membership queries to obtain a sample $\varepsilon$-close to $\pi$ in TV is $\widetilde{\mathcal{O}}(Md^2 D^2 \log\frac{1}{\varepsilon})$, which now has a poly-logarithmic dependence on $1/\varepsilon$.

*Remark* 3 (Backward heat flow analysis of Speedy walk). Consider a Gaussian version of Speedy walk, whose one-step corresponds to $x_{i+1} \sim \mathcal{N}(x_i, hI_d)|_{\mathcal{K}}$, and this transition kernel exactly matches integrating (BH) for time $h$. Thus, $\nu Q_h^{\pi^X, h} = \nu$ due to the stationarity of $\nu$ under Speedy walk, where $Q_h^{\pi^X, h}$ is the transition kernel defined by the backward heat flow for time $h$ that reverses $\pi^X * \mathcal{N}(0, hI_d)$ to $\pi^X$. Hence, if we can control the LSI/PI constants of $\nu$ along the backward heat-flow's trajectory, then we could directly analyze Speedy walk by emulating computations in Lemma 4.

## D  Functional inequalities

We provide full details on functional inequalities omitted in Appendix B.1. We use $\mu$ and $\mu_{\mathsf{LC}}$ to denote a probability measure and log-concave probability measure over $\mathbb{R}^d$, respectively.

**Cheeger and PI constants.** The *Cheeger isoperimetric constant* $C_{\mathsf{Ch}}(\mu)$ measures how large surface area a measurable subset with larger volume has, defined by

$$C_{\mathsf{Ch}}(\mu) := \inf_{S \subset \mathbb{R}^d} \frac{\mu^+(S)}{\min(\mu(S), \mu(S^c))}\,,$$

where the infimum is taken over all measurable subsets $S$, and $\mu^+(S)$ is the Minkowski content of $S$ under $\mu$ defined as, for $S^\varepsilon := \{x \in X : d(x, S) < \varepsilon\}$,

$$\mu^+(S) := \liminf_{\varepsilon \to 0} \frac{\mu(S^\varepsilon) - \mu(S)}{\varepsilon}\,.$$

[62] established $C_{\mathsf{PI}}(\mu) \lesssim C_{\mathsf{Ch}}^{-2}(\mu)$[5], and then [33] showed that for covariance matrix $\Sigma_\mu := \mathbb{E}_\mu[(\cdot - \mathbb{E}_\mu X)(\cdot - \mathbb{E}_\mu X)^\mathsf{T}]$,

$$C_{\mathsf{Ch}}(\mu_{\mathsf{LC}}) \gtrsim \frac{1}{(\mathbb{E}_{\mu_{\mathsf{LC}}}[\|X - \mathbb{E}_{\mu_{\mathsf{LC}}} X\|^2])^{1/2}} = \frac{1}{(\mathrm{tr}\,\Sigma_{\mu_{\mathsf{LC}}})^{1/2}}\,. \tag{D.1}$$

---

[5]The opposite direction $C_{\mathsf{PI}}(\mu_{\mathsf{LC}}) \gtrsim C_{\mathsf{Ch}}^{-2}(\mu_{\mathsf{LC}})$ also holds for log-concave distributions due to [63], while $C_{\mathsf{PI}}(\mu) \gtrsim C_{\mathsf{Ch}}^{-2}(\mu)/d$ for general distributions due to [64].

This immediately leads to $C_{\mathsf{PI}}(\pi) \lesssim (\mathbb{E}_\pi[\|X - \mathbb{E}_\pi X\|^2])^{1/2} \leq D^2$ for the uniform distribution $\pi$ over a convex body $\mathcal{K}$ with diameter $D > 0$.

Kannan et al. proposed the *KLS conjecture* in the same paper, which says that for the spectral norm $\|\cdot\|_2$,

$$C_{\mathsf{Ch}}(\mu_{\mathsf{LS}}) \gtrsim \frac{1}{\|\Sigma_{\mu_{\mathsf{LS}}}\|_2^{1/2}}.$$

While the original result in [33] ensures $C_{\mathsf{Ch}} \gtrsim d^{-1/2}$ for an isotropic log-concave distribution (due to $\Sigma = I_d$), this conjecture indeed claims $C_{\mathsf{Ch}} \gtrsim 1$ for such case. Following a line of work [37, 55, 65, 34], the current bound is

$$C_{\mathsf{Ch}}(\mu_{\mathsf{LS}}) \gtrsim \frac{(\log d)^{-1/2}}{\|\Sigma_{\mu_{\mathsf{LS}}}\|_2^{1/2}},$$

which implies that $C_{\mathsf{PI}}(\pi) \lesssim \log d$ when $\pi$ is isotropic for convex $\mathcal{K}$.

**Log-Cheeger and LSI constants.** Just as the Cheeger and PI constants are related above, there are known connections between LSI and *log-Cheeger* constants. The log-Cheeger constant $C_{\mathsf{logCh}}(\mu)$ of a distribution $\mu \in \mathcal{P}(\mathbb{R}^d)$ is defined as

$$C_{\mathsf{logCh}}(\mu) := \inf_{S \subset \mathbb{R}^d : \mu(S) \leq \frac{1}{2}} \frac{\mu^+(S)}{\mu(S)\sqrt{\log \frac{1}{\mu(S)}}}.$$

[64] established that $C_{\mathsf{LSI}}(\mu) \lesssim C_{\mathsf{logCh}}^{-2}(\mu)$[6], and [61] showed that any log-concave distributions with support of diameter $D > 0$ satisfy $C_{\mathsf{logCh}}(\mu_{\mathsf{LS}}) \gtrsim D^{-1}$. Later in 2016, [37] improved this to $C_{\mathsf{logCh}}(\mu_{\mathsf{LS}}) \gtrsim D^{-1/2}$ under isotropy. Therefore, for convex $\mathcal{K}$, it follows that $C_{\mathsf{LSI}}(\pi) \lesssim D^2$ and that $C_{\mathsf{LSI}}(\pi) \lesssim D$ if $\pi$ is isotropic.

# E  The Wasserstein geometry

We present additional technical background on the Wasserstein geometry and Markov semigroup theory. Interested readers can refer to [66, 67, 28] for standard references on Wasserstein spaces and applications to sampling.

**Wasserstein gradient.** Let $\mathcal{P}_{2,\mathrm{ac}}(\mathbb{R}^d)$ be the space of probability measures admitting densities on $\mathbb{R}^d$ with finite second moment. Although there are many ways to metrize $\mathcal{P}_{2,\mathrm{ac}}(\mathbb{R}^d)$, the geometry induced by the Wasserstein-2 distance $\mathcal{W}_2$ is a particularly useful structure for analysis.

Under the $\mathcal{W}_2$-geometry, one can define a "gradient" of a functional defined over $\mathcal{P}_{2,\mathrm{ac}}(\mathbb{R}^d)$. Specifically, for a functional $\mathcal{F} : \mathcal{P}_{2,\mathrm{ac}}(\mathbb{R}^d) \to \mathbb{R} \cup \{\infty\}$, the *Wasserstein gradient* of $\mathcal{F}$ at $\mu \in \mathcal{P}_{2,\mathrm{ac}}(\mathbb{R}^d)$ is defined as $\nabla_{\mathcal{W}_2}\mathcal{F}(\mu) = \nabla(\delta\mathcal{F})(\mu) \in L^2(\mu)$, where $\nabla$ is the standard gradient and $\delta\mathcal{F}$ is the first variation of $\mathcal{F}$[7]. Equipped with this $\mathcal{W}_2$-gradient, one can define the *Wasserstein gradient flow* of $\mathcal{F}$ that describes the evolution of a measure $\{\mu_t\}_{t \geq 0}$, from some initial measure $\mu_0$, as follows:

$$\partial_t \mu_t = \mathrm{div}\big(\mu_t \nabla_{\mathcal{W}_2}\mathcal{F}(\mu_t)\big).$$

More generally, we can identify the Wasserstein "velocity" for some measure $\mu_t$ as $v_t$ if the time derivative of $\mu_t$ can be written in the form

$$\partial_t \mu_t = -\,\mathrm{div}(\mu_t v_t).$$

Under this identification, the time derivative of a functional $\mathcal{F}$ on $\mathcal{P}_{2,\mathrm{ac}}(\mathbb{R}^d)$ with smooth Wasserstein gradient under these dynamics can be written as

$$\partial_t \mathcal{F}(\mu_t) = \mathbb{E}_{\mu_t}\langle \nabla_{\mathcal{W}_2}\mathcal{F}(\mu_t), v_t\rangle,$$

---

[6]The opposite direction holds under dimension-scaling due to [64]: $C_{\mathsf{LSI}}(\mu) \gtrsim C_{\mathsf{logCh}}^{-2}(\mu)/d$.

[7]The first variation can be defined, for any measures $\nu_0, \nu_1 \in \mathcal{P}_{2,\mathrm{ac}}(\mathbb{R}^d)$, as $\lim_{t \downarrow 0} \frac{\mathcal{F}((1-t)\nu_0 + t\nu_1) - \mathcal{F}(\nu_0)}{t} = \langle \delta\mathcal{F}(\nu_0), \nu_1 - \nu_0\rangle$. This definition is unique up to an additive constant, which is irrelevant as we are only concerned with its gradient.

when $v_t \in \overline{\{\nabla \psi : \psi \in C_c^\infty(\mathbb{R}^d)\}}^{L^2(\mu_t)}$, where $\overline{\{\cdot\}}^{L^2(\mu_t)}$ denotes the closure of a set with respect to $L^2(\mu_t)$. This is the appropriate notion of tangent space in this geometry.

For instance, when we take the functional to be the entropy of the measure, $\mathcal{H}(\mu) := \frac{1}{2} \int \mu \log \mu$, one can verify $\nabla_{\mathcal{W}_2} \mathcal{H}(\mu) = \frac{1}{2} \nabla \log \mu$. The heat flow equation can be written as $\partial_t \mu_t = \frac{1}{2} \Delta \mu_t = \frac{1}{2} \operatorname{div}(\nabla \mu_t) = \frac{1}{2} \operatorname{div}(\mu_t \nabla \log \mu_t)$, which indicates that the velocity of measures $\mu_t$ under the heat flow is $v_t = -\frac{1}{2} \nabla \log \mu_t$. Hence, we can notice that $\nabla_{\mathcal{W}_2} \mathcal{H}(\mu_t) = -v_t$, and thus recover the heat flow as the Wasserstein gradient flow of the entropy of the measure.

**Fokker-Planck equation and time-reversal of SDE.**    Consider a stochastic differential equation $(X_t)$ given by

$$\mathrm{d}X_t = -a_t(X_t)\,\mathrm{d}t + \mathrm{d}B_t \qquad \text{with } X_0 \sim \mu_0 \,. \tag{E.1}$$

It is well known that measures $\mu_t$ described by

$$\partial_t \mu_t = \operatorname{div}(\mu_t a_t) + \frac{1}{2}\Delta \mu_t \,, \tag{E.2}$$

correspond to $\operatorname{law}(X_t)$. In this context, (E.2) is referred to as the *Fokker-Planck equation* corresponding to (E.1).

From this equation, one can deduce the Fokker-Planck equation of the time reversal $\mu_t^{\leftarrow} := \mu_{T-t}$:

$$\partial_t \mu_t^{\leftarrow} = -\operatorname{div}(\mu_t^{\leftarrow} a_{T-t}) - \frac{1}{2}\Delta \mu_t^{\leftarrow} = -\operatorname{div}\big(\mu_t^{\leftarrow}(a_{T-t} + \nabla \log \mu_{T-t})\big) + \frac{1}{2}\Delta \mu_t^{\leftarrow}$$

In particular, this describes the evolution of $\operatorname{law}(X_t)$ of the stochastic differential equation:

$$\mathrm{d}X_t = \big(a_{T-t}(X_t) + \nabla \log \mu_{T-t}(X_t)\big)\,\mathrm{d}t + \mathrm{d}B_t \qquad \text{with } X_0 \sim \mu_0^{\leftarrow} = \mu_T \,. \tag{E.3}$$

While the law of this process will give $\mu_T^{\leftarrow} = \mu_0$ at time $T$, it is also true that it will give $\mu_{0|T}(\cdot|z)$ if one starts (E.3) at $X_0 = z$. This is a subtle fact, whose justification requires the introduction of a tool called *Doob's h-transform*. The presentation of this subject is beyond the scope of this paper, and we refer interested readers to [53] as a reference to its application in this context.

