# OpenReview forum: "In-and-Out: Algorithmic Diffusion for Sampling Convex Bodies"
_NeurIPS.cc/2024/Conference — NeurIPS 2024 spotlight_

### Official Review · Reviewer_Jx5G · 2024-07-02

**Soundness:** 3
**Presentation:** 3
**Contribution:** 3
**Rating:** 6
**Confidence:** 3

**Summary:**

In this paper, the authors consider the problem of uniform sampling from general convex body. The algorithm works by augmenting the state space followed by performing alternative Gibbs sampling, where one of the inner steps is implemented by rejection sampling. Non-asymptotic end-to-end bounds on the mixing time in Renyi divergence under warmness assumption is established. The analysis takes the view of the two updates as forward / backward heat flows, building on existing proximal sampler work, therefore offers a clean stochastic process perspective on the algorithm. Various connections of the proposed algorithm to ball walk and speedy walk are also discussed.

**Strengths:**

The paper is clearly written and well-structured. Relevant related works are adequately surveyed with the topic being of great interest to many problems in machine learning and computational statistics. While the results and analysis build on the existing framework of proximal sampler for log-concave sampling, I think there are contributions in the paper to this new setting that's worth sharing with the community. The approach is conceptually simple and offers a fresh perspective on constrained sampling.

**Weaknesses:**

I have listed some questions in the section below. My main concern is that while I understand the authors provide guarantee in a stronger metric compared to existing algorithm / analysis, given that the query complexity of In-and-Out matches that of some previous work, and the methods are similar in some regard (for example, the second step is conceptually similar to a projection step), it's unclear to me what the practical advantage of the proposed method may be.

Minor comment: Some kind of table putting together all the mentioned rates under different assumptions / metrics alongside related previous results in the literature will help the reader digest things a bit better.

**Questions:**

- I haven't given this a careful thought - does the result follow from the proximal sampler result by taking some suitable limit? (since the target is log-concave here and "discretization" isn't an issue for Gibbs sampler) Since I'm confused as to why warmness would show up here but not in the original proximal sampler?

- I didn't quite follow Line 146-149 - how would In-and-Out compare to projection-based method?


- Line 234-236 the comment about being lazy: In the proposed algorithm, if it doesn't make a proper move (i.e., declares "Failure"), wouldn't that be equivalent to a "lazy" step where things don't move? Or maybe another way to say this is - how should one take ergodic averages along the chain for computing expectation of some observable? It's not entirely clear from first reading.


- How does one obtain a warm start in practice?

**Limitations:**

Yes the work is mostly theoretical.

---

> ### Author Rebuttal · Authors · 2024-08-03
>
> $\textbf{What practical advantage of the method?}$
>
> The main advantage of our method is the simplicity of the algorithm and the analysis with provable guarantees in commonly used probabilistic metrics, more general than previously known. Not only is our algorithm much simpler to analyze, but it also provides some tangible improvements (such as improving the guarantee from $\mathsf{TV}$ to $\mathsf{KL}$ and $\mathcal{R}_q$); it also shows a clear and direct connection between isoperimetry and convergence.
> In practice, we might expect comparable performance between all these approaches, although this would depend on the details of the implementation.
>
> $\textbf{Does the result follow from the proximal sampler by taking some suitable limit}$
>
> As demonstrated in Lemma 12 and Theorem 3, the mixing guarantee of our sampler (i.e., how many outer loops are needed) does follow from a limiting argument. However, what matters in the end is a bound on query complexity (not just mixing rate), which essentially requires us to bound the number of rejections throughout backward steps.
>
> The approach required for the constrained case is not comparable to the unconstrained case. Previous work on proximal sampling works under a well-conditioned setting without hard constraints, where the number of trials for rejection sampling (for the backward step) is always $\mathcal{O}(1)$ regardless of the forward step. However, in the presence of constraints, this type of analysis for the backward step is no longer possible. Analyzing proximal-type sampling for this setting has been a well-known open problem. Instead, we carried out the analysis for the backward step in a more careful way. As a result, there are a number of new ideas in the rejection analysis.
>
> $\textbf{Line 146-149: How would INO compare to projection-based methods}$
>
> In general, a projection oracle is stronger than a membership oracle, and its implementation using a membership oracle requires $O(d^2)$ membership calls per projection in the worst case.
> One can ask instead what happens if we assume a projection oracle which is roughly the same cost as a membership oracle. In this case, leveraging projection might be faster, but this is a non-trivial problem. We leave this for future work.
>
> $\textbf{Failure = Lazy? How should one take ergodic averages along the chain for computing expectation of some observable?}$
>
> We thank the reviewer for this question. In our view, failure of our chain is not the same as taking lazy steps in e.g. Ball walk. Lazy chains have a 1/2 chance to remain stationary at each instant, potentially doubling the iterations needed for convergence.
> By contrast, our algorithm has an arbitrarily small failure probability (moreover, the mixing guarantee has only a poly-log dependence $\text{polylog}(1/\eta)$). In the event of failure, we restart the algorithm until we succeed, which does not introduce any bias.
>
> Regarding the second question, it is still possible to take ergodic averages along the chain (assuming you pick the failure probability at each iteration to be sufficiently small so that your run rarely fails during the entire horizon of your run). This is because our Markov chain still has a spectral gap, and so the same techniques used to obtain guarantees for ergodic averages continue to work in our case.
>
> $\textbf{How to obtain a warm start in practice}$
>
> Obtaining warm-starts in practice is non-trivial, and requires some type of annealing algorithm in general. Indeed, warm-start generation has been studied for more than two decades in theoretical computer science. See e.g., [1, 2] for how this can be done. For simplicity, we assume that a warm start is provided in our algorithm, although one can use the guarantees in [1, 2] to provide a rigorous justification.
>
> ***
>
> [1] Gaussian Cooling and $O^*(n^3)$ Algorithms for Volume and Gaussian Volume, Ben Cousins and Santosh S. Vempala.
>
> [2] Reducing Isotropy and Volume to KLS: An $O(n^3\psi^2)$ Volume Algorithm, Jia et al.

---

> > ### Comment · Reviewer_Jx5G · 2024-08-09
> >
> > Thanks for clarifying and the explanation. I do think the paper makes contribution that the community would benefit from seeing.
> >
> > My slight reservation comes from the fact that most of the contribution comes from analyzing the rejection sampling part, in which the M-warmness condition is doing most of the heavy-lifting, so perhaps in some sense the stated results are expected.

---

> ### Author Response · Authors · 2024-08-12
>
> We are unclear on the reviewer's "reservation". The problem of sampling constrained convex bodies has been widely studied for decades and so far did not have guarantees beyond TV distance; moreover, in recent years, it has been a well-known open problem to see if diffusion-based methods could be used to obtain a polytime algorithm. Our main results show stronger guarantees for the classical problem using a simple and clean diffusion approach. The resulting analysis brings together several well-known analysis components, along with some extensions (to the constrained setting) and some new ideas (rejection analysis). We feel that the fact the solution is relatively simple and does not need substantial technical sophistication is an attractive feature.

---

### Official Review · Reviewer_V3VH · 2024-07-11

**Soundness:** 4
**Presentation:** 3
**Contribution:** 3
**Rating:** 8
**Confidence:** 5

**Summary:**

The paper presents a novel random walk algorithm for uniform sampling of high-dimensional convex bodies that provides improved runtime complexity and guarantees on the result, especially with respect to Rényi divergence.
Sampling high-dimensional convex bodies, a fundamental problem in algorithm theory with numerous applications in scientific computing, systems biology, differential privacy, and (to a lesser extent) machine learning. All the samplers known so far rely on Markov chains and most of the time, convergence analysis depends on limiting the conductivity of the associated chain, which in turn controls the mixing rate.

The algorithm alternates in and out moves, which is a kind of modification of ball walk, avoiding MH step. The theoretical analysis shows that the method contracts the distribution towards the target distribution at a rate that is influenced by the isoperimetric properties of the convex body.
The results show the effectiveness of the new algorithm compared to traditional methods such as the ball-walk and hit-and-run algorithms. The "in-and-out" method shows superior performance, especially in high-dimensional environments, due to its direct reduction to isoperimetric constants.

**Strengths:**

- Introduction of the "in-and-out" algorithm for uniform sampling that uses a heat flow approach.
-  stronger guarantees  in terms of Rényi divergence, which includes other divergence measures such as total variation (TV), Wasserstein (W2), Kullback-Leibler (KL) and chi-squared (χ2).
- analysis of mixing rates from a heat flow perspective, providing new insights and extending known results for the unconstrained domain.
Convergence rate is shown to be determined by functional inequalities such as Poincaré (PI) and Log-Sobolev inequalities (LSI-I).
Iteration complexity: For isotropic distributions, the algorithm achieves convergence with a polynomial number of iterations depending on the dimension and the desired accuracy.

**Weaknesses:**

- I don't see any major weaknesses. The paper is easy to read, the main stages of the analysis are outlined in the text (I didn't have time to check the appendices) and are interesting. I regret the absence of a numerical section to compare the interest of the In-and-Out method on practical examples [comparing it to existing algorithms].

**Questions:**

- Can you provide more details on the initialization process and parameter selection (e.g., h, N) for the algorithm?
- Could you elaborate on how the functional isoperimetric constants influence the convergence rate of the algorithm for specific geometry (polytopes, ellipsoids, etc...) ?
- How can the algorithm be extended to sample from general log-concave distributions restricted to convex bodies or other non-log-concave distributions satisfying isoperimetric inequalities? [is there a "general" idea that can be preserved? - replacing the "hard" Metropolis filter by and In-and-Out mechanism ?]
-

**Limitations:**

yes

---

> ### Author Rebuttal · Authors · 2024-08-03
>
> $\textbf{Initialization process and parameter selection}$
>
> - Initialization process: We assume that the initial start is $M$-warm in this work. Obtaining this warm-start is non-trivial, and requires some type of annealing algorithm in general. Warm-start generation has been studied for more than two decades in theoretical computer science. See e.g. [1] for how this can be done.
> - Parameter selection: The details of $N, h, \eta$ are given in the proof (see Lemma 3 and 14 for example). We will make this more clear in the initial Theorem statement as well. Note that we do not track the absolute constants hidden in big-O notation, but this can be obtained from a careful reading of the proof; our analysis was not optimal with respect to numerical constants, which can likely be sharpened with a more precise argument.
>
> $\textbf{How FI parameters affect the convergence rate for specific geometry}$
>
> The Poincare constant in general is on the same order as the maximum eigenvalue of the covariance, while the log-Sobolev constant is bounded by the squared diameter of the convex body (please refer to Appendix D).
> However, for a more structured body, it is not clear what the sharpest possible constant is for these functional inequalities. In general, we expect that it would be very difficult to estimate unless the body is something simple like an $\ell_p$ ball.
> We can also obtain the constants under a linear map $T: x \mapsto Ax$ by multiplying the constants by $||A||^2$.
>
> $\textbf{Extension to a general setting}$
>
> We thank the reviewer for their insightful suggestion. While our framework could potentially accommodate it, the rates after incorporating a first-order oracle for sampling $e^{-f}1_K$ are not immediately clear from our analysis and would take some more effort. We leave this for future work.
>
> ***
>
> [1] Gaussian Cooling and $O^*(n^3)$ Algorithms for Volume and Gaussian Volume, Ben Cousins and Santosh S. Vempala.

---

> > ### Comment · Reviewer_V3VH · 2024-08-13
> >
> > This is a good paper. I will keep my score !

---

### Official Review · Reviewer_T79w · 2024-07-12

**Soundness:** 3
**Presentation:** 3
**Contribution:** 4
**Rating:** 7
**Confidence:** 4

**Summary:**

The paper addresses the fundamental problem of uniformly sampling high-dimensional convex bodies. The main contribution is the proposal of the In-and-Out algorithm, analyzed within the framework of the proximal sampling scheme. Using existing analyses from the literature, the paper derives strong results. Additionally, the authors discuss classical methods for constrained sampling and diffusion-based or proximal methods for unconstrained sampling.

**Strengths:**

- Overall, I like this paper. It proposes a new method and offers valuable insights. Additionally, the paper achieves strong results with straightforward proofs.
- The paper is well-written and easy to follow.

**Weaknesses:**

- The analysis techniques used in the paper already exist in the literature, which limits the technical contribution.
-The paper does not provide a comparison of iteration/query complexity with existing works.

**Questions:**

- Is it possible to extend these results to the log-concave setting?
- The membership oracle used is standard. What if we are given the polytope constraint explicitly?

---

> ### Author Rebuttal · Authors · 2024-08-03
>
> $\textbf{Techniques are already existing}$
>
> As noted by Reviewer W5VJ, our paper is not “just” putting together known components. Previous work on proximal sampling works under a well-conditioned setting without hard constraints, where the number of trials for rejection sampling (for the backward step) is always $\mathcal{O}(1)$ regardless of the forward step. However, in the presence of constraints, this type of analysis for the backward step is no longer possible. Analyzing proximal-type sampling for this setting has been a well-known open problem. Instead, we carried out the analysis for the backward step in a more careful way. As a result, there are a number of new ideas in the rejection analysis.
>
> $\textbf{Extension to a general setting}$
>
> We thank the reviewer for their insightful suggestion. While our framework could potentially accommodate it, the rates after incorporating a first-order oracle for sampling $e^{-f}1_K$ are not immediately clear from our analysis and would take some more effort. We leave this for future work.
>
> $\textbf{Polytope constraint?}$
>
> There are several ways of leveraging the explicit structure of the log-concave sampling problem. For instance, one can easily implement a projection oracle for polytope constraints (which is stronger than a membership one), so one may consider a projection-based algorithm. Another approach for exploiting the structure is to use barrier-related information. In general, a convex constraint admits a self-concordant barrier $\phi$, and samplers with the local metric given by the Hessian $\nabla^2 \phi$ are provably fast and practical (due to condition-number independence, for instance).

---

> > ### Comment · Reviewer_T79w · 2024-08-13
> >
> > Thank you very much for your detailed response.

---

### Official Review · Reviewer_W5VJ · 2024-07-15

**Soundness:** 4
**Presentation:** 4
**Contribution:** 4
**Rating:** 8
**Confidence:** 4

**Summary:**

This work presents a new algorithm, called In-and-Out, for sampling from the uniform distribution over a convex subset $K$ of $R^d$ that comes with stronger guarantees than previous algorithms. The proposed algorithm is an instantiation of the Proximal Sampler (PS), an abstract sampling algorithm which was recently shown to have very strong guarantees (Chen et al., 2022 [27]). The PS was previously considered for sampling from $\propto e^{-f} dx$ given first-order oracle access to $f$, and was typically implemented using a form of rejection sampling; the guarantees in [27] are stated purely in terms of the Poincare Inequality (PI) or Log-Sobolev Inequality (LSI) constant of the target. In the setting of this work, the target is $\propto 1_K$ the uniform distribution on a set $K$ for which we have membership oracle access, and the implementation also uses (another form of) rejection sampling. The guarantees presented in this work follow from the analysis from [27] (§B.2) and a regularity lemma (Lemma 12), from bounds on the PI and LSI constants of uniform distributions known from various recent works (Lemma 9 and §D), and from fine bounds on the failure probability in rejection sampling (§B.3).

**Strengths:**

From the point of view of the problem of sampling convex bodies, the contributions of this paper are outstanding. The results appear much stronger than previously known guarantees (but I am not familiar with the literature on this problem, so I can only trust the authors's discussion of the related work.) It illustrates that the diffusion approach to sampling can yield strong results on a problem where geometric approaches are perhaps more natural.

From the point of view of algorithmic diffusion, this paper is not "just" a piecing together of several known components, as the analysis of the failure probability of the rejection step is not at all obvious. I found the concise rewriting of the analysis of [27] (Part I of §B.2) quite appreciable as well.

**Weaknesses:**

- No application is presented or discussed. Usually for theoretical works such as this one, the value of the contribution lies in the analysis technique, with the hope that it will allow to eventually obtain guarantees for future applications. But the applications that naturally come to my mind are cases which the results of [27] already cover.

Some minor suggestions:
- The relation between Thms 1, 2, 3 is a little bit confusing, as only Thms 2, 3 talk about PI and LSI constants. Perhaps the presentation of the results could be made clearer by moving Lemma 9 to the end of §2, or at least making a reference to it there.
- I would suggest mentioning already in the abstract or the introduction that In-and-Out is an instance of PS. This would be fairer w.r.t. prior work in my opnion.
- The fact that Part I of §B.2 is a restatement of the analysis of [27] should be clarified (the terms "revisit" and "review" currently used on lines 563, 569 are not completely clear).
- On lines 139-141, you mention that the time-reversal SDE has the property that it is also a reversal "pointwise", i.e, conditional on the endpoint. But I did not see where this fact is used, as in §B.2 only the reversal of the heat flow at the PDE level is used.

**Questions:**

- Regarding the first point in "Weaknesses": what are some application cases in machine learning where your work applies?
- Can the In-and-Out algorithm be extended to sampling from a target $\propto e^{-f} 1_K$ where $f$ is smooth on $R^d$ given first-order access to $f$ and membership oracle access to $K$? Specifically the rejection sampling step and its failure probability analysis (since I expect the analysis of the abstract algorithm, PS, is unchanged)
- On line 175, you mention a better query complexity if $K$ is near-isotropic, but Corollary 1 just below is about the exactly-isotropic case. What does line 175 refer to? (Is it Thm 5? if yes it should be mentioned in the main text)
- Out of curiosity: are there ways to rescale a convex body to be near-isotropic, given membership oracle access? This could give a preprocessing step which could improve the query complexity you report.

**Limitations:**

The authors have adequately addressed the limitations.

---

> ### Author Rebuttal · Authors · 2024-08-03
>
> $\textbf{Application}$
>
> Our work mainly aims to provide a new algorithmic/analytic framework for the uniform sampling problem under the membership oracle model. The applications of this problem are already widespread and well-known, therefore we do not feel the need to propose any new applications. Instead, we note that this problem is used in a number of fundamental settings: volume computation for convex bodies; as a core subroutine in general convex body sampling; metabolic flux sampling in systems biology; Bayesian inference in statistics; and other domains of scientific computing. Thus, new results for the core algorithmic problem would immediately imply theoretical improvements for all these settings.
>
> $\textbf{Extension to potentials}$
>
> We thank the reviewer for their insightful suggestion. While our framework could potentially accommodate it, the rates after incorporating a first-order oracle for sampling $e^{-f}1_K$ are not immediately clear from our analysis and would take some more effort. We leave this for future work.
>
> $\textbf{Confusion around Line 175}$
>
> We apologize for the confusion. Corollary 1 holds for the near-isotropic case as well, since the operator norm of the covariance is $\mathcal{O}(1)$.
>
> $\textbf{How to make it isotropic}$
>
> The question of obtaining an isotropic position has been studied for more than two decades in theoretical computer science. For references, please consult [1, 2] for an idea of how this can be done. In summary, their approach is to draw a few samples from the body and then obtain a rough estimate of the covariance $T$. Then, applying a proper linear map $T^{-1/2}$ to the body will reduce the skewness. Repeating these steps $\mathcal{O}(\log d)$ many times (along with some type of annealing algorithm) ensures that a transformed body is nearly isotropic with high probability. Although we had mentioned this procedure in our related works, we will make a comment earlier in our paper for clarity. We thank the reviewer for their question.
>
> $\textbf{Regarding Line 139-141 (time-reversal of SDE)}$
>
> The final paragraph of Appendix E discusses why the pointwise reversal property is needed. In particular, it allows us to claim that given a starting point $z$, the reverse SDE from such a starting point has distribution $\pi^{X|Y=y}$, a property used in the initial paragraphs of Appendix B. This will be clarified in our revision.
>
> ***
>
> [1] Gaussian Cooling and $O^*(n^3)$ Algorithms for Volume and Gaussian Volume, Ben Cousins and Santosh S. Vempala.
>
> [2] Reducing Isotropy and Volume to KLS: An $O(n^3\psi^2)$ Volume Algorithm, Jia et al.

---

> > ### Comment · Reviewer_W5VJ · 2024-08-08
> >
> > Re Applications: I would still recommend adding a few words on practical applications where the membership oracle model is the way to go, if you know of some and they are not too complicated to explain. Otherwise that's ok.
> >
> > Re Line 175: Thank you for clarifying this in the final version.
> >
> > Re Making it isotropic: My bad, I had somehow missed this. But mentioning this procedure earlier in the paper would indeed be beneficial.
> >
> > Re potentials and pointwise time-reversal: fair enough.
> >
> > I maintain my positive rating. Congratulations for a very nice paper :)

---

### Author Rebuttal · Authors · 2024-08-03

We thank all the reviewers for their time and detailed comments. We respond to specific points below. We note at the outset that main high-level contributions are that (a) we provide the first guarantees for KL and Renyi divergences and (b) we directly relate the convergence rates to classical isoperimetric constants of the target distribution.

We make extensive simplifications to the convergence analysis through our approach, which also exposes a clear relationship between the complexity and the isoperimetry of the target distribution. This immediately gives improvements to guarantees for this problem, with rate: $\mathcal{O}(qd^2 M \Lambda \log 1/\varepsilon)$ in general, where $\Lambda$ is the maximum eigenvalue of the covariance matrix and the convergence is in $q$-Renyi.

$\textbf{Rate Comparison}$: One question raised by multiple reviewers concerns the relationship of the rates obtained in our work, as compared to the best known prior results in this setting. Below, we highlight the main results in constrained uniform sampling under the membership oracle before this work:
- Ball walk: the rate is $\mathcal{O}(Md^2 \psi_{\text{Cheeger}} \log 1/\varepsilon)$ in TV. It relies on a conductance argument.
- Hit-and-run: the rate is $\mathcal{O}(d^2 R^2 \log M/\varepsilon)$ to mix in $\chi^2$, where $R^2$ is the trace of the covariance. It also relies on a conductance argument.

These rates were given in the related work section of our paper. In our updated draft, we will also state them immediately after our theorems.

---

### Decision · Program_Chairs · 2024-09-25

**Decision:**

Accept (spotlight)

**Comment:**

The paper gives a new algorithm for the classical problem of sampling from a convex body. The new algorithm uses a different approach from many previous works for this problem, yielding convergence guarantees in Renyi divergences, which are more general than previous results. All reviewers support the paper to varying degrees with several putting it at strong accept.

The paper is purely theoretical and does not include any numerical experiment comparing with existing methods.